# Simulating Marine Isotope Stage 7 with a coupled climate-ice sheet model

Dipayan Choudhury[1,2], Axel Timmermann[1,2], Fabian Schloesser[3], Malte Heinemann[4] and David Pollard[5]

[1]Center for Climate Physics, Institute for Basic Science (IBS), Busan 46241, South Korea
[2]Pusan National University, Busan 46241, South Korea
[3]International Pacific Research Center, University of Hawaii at Manoa, Honolulu, HI 96822, USA
[4]Institute of Geosciences, Kiel University, 24118, Kiel, Germany
[5]Earth and Environmental Systems Institute, Pennsylvania State University, Pennsylvania 16802, USA

*Correspondence to*: Dipayan Choudhury (dipayanc@pusan.ac.kr)

**Abstract.** It is widely accepted that orbital variations are responsible for the generation of glacial cycles during the late Pleistocene. However, the relative contributions of the orbital forcing compared to $CO_2$ variations and other feedback mechanisms causing the waxing and waning of ice-sheets have not been fully understood. Testing theories of ice-ages beyond statistical inferences, requires numerical modeling experiments that capture key features of glacial transitions. Here, we focus on the glacial build-up from Marine Isotope Stage (MIS) 7 to 6 covering the period from 240-170 ka (thousand years before
present). This transition from interglacial to glacial conditions includes one of the fastest Pleistocene glaciation/deglaciation events which occurred during MIS 7e-7d-7c (236-218ka).  Using a newly developed three-dimensional coupled atmosphere-ocean-vegetation-ice-sheet model (LOVECLIP), we simulate the transient evolution of northern and southern hemisphere ice-sheets during the MIS 7-6 period in response to orbital and greenhouse-gas forcing. For a range of model parameters, the simulations capture the evolution of global ice volume well within the range of reconstructions. Over the MIS 7-6 period, it is
demonstrated that glacial inceptions are more sensitive to orbital variations, whereas terminations from deep glacial conditions need both orbital and greenhouse gas forcings to work in unison. For some parameter values, the coupled model also exhibits a critical North American ice sheet configuration, beyond which a stationary wave – ice-sheet topography feedback can trigger an unabated and unrealistic ice-sheet growth. The strong parameter sensitivity found in this study originates from the fact that delicate mass imbalances, as well as errors, are integrated during a transient simulation for thousands of years. This poses a
general challenge for transient coupled climate-ice sheet modeling, with such coupled paleo-simulations providing opportunities to constrain such parameters.

## 1 Introduction

Earth's climate over the past one million years (Late Quaternary) is characterized by glacial/interglacial cycles representing cold/warm periods, transitioning in timescales of around 80,000-120,000 years. These transitions correspond to global sea level changes of up to 130m (Fig. 1b) (Waelbroeck et al., 2002;Lisiecki and Raymo, 2005;Bintanja et al., 2005). Simulating these massive reorganizations of earth's climate using earth system models of varying complexity is an active area of research. By comparing such simulations with paleoclimate data, we can evaluate the fidelity of these climate models, as well as refine our understanding of the underlying sensitivities and feedbacks to a variety of forcings. One of the main obstacles in simulating variability on orbital timescales is the fact that ice-sheets are slow integrators of small imbalances between ablation and accumulation, which correspond to an average of 1.3mm/year global sea level equivalent during the build-up phase but can exceed 10mm/year for instance during the Last Glacial Maximum (LGM, 21ka). In order to simulate an entire glacial/interglacial cycle, model errors can accumulate for thousands of years and potential multiple equilibria of the fully coupled system can create further complications. Simulating a transient climate "trajectory" realistically is an even bigger modeling and computational challenge than simulating climate snapshots realistically, such as for the LGM (Yoshimori et al., 2002;Lunt et al., 2013;Colleoni et al., 2014a;Rachmayani et al., 2016).

Most efforts so far, with the notable exception of Ziemen et al. (2019), have used earth system models of intermediate complexity, EMICs, (Ganopolski and Brovkin, 2017;Ganopolski et al., 2010;Stap et al., 2014;Vizcaino et al., 2015;Calov et al., 2005;Heinemann et al., 2014;Willeit et al., 2019) or ice-sheet models (ISMs) coupled to statistical relationships, based on a set of coupled general circulation model (CGCM) timeslice runs (Abe-Ouchi et al., 2013;Colleoni et al., 2014b) to simulate the transient evolution of the coupled atmosphere-ocean-ice-sheet system. Bi-directional coupling between climate components and the ice-sheets, typically not captured in offline ice-sheet simulations (Born et al., 2010;Dolan et al., 2015;Koenig et al., 2015), is crucial in representing important feedbacks such as the ice albedo (Abe-Ouchi et al., 2013), elevation-desertification (Yamagishi et al., 2005) and the stationary wave – ice-sheet (Roe and Lindzen, 2001) feedbacks. Furthermore, it has been argued that the interaction between ice-sheets and ocean circulation (Timmermann et al., 2010;Rahmstorf, 2002;Knutti et al., 2004), and the resulting effects on the marine carbon cycle (Gildor and Tziperman, 2000;Menviel et al., 2012;Stein et al., 2020) can play a first-order role in shaping the climate evolution of the Quaternary on millennial and orbital timescales.

Glacial inceptions from warm mean states (interglacials) to cold mean states (glacials) over relatively short periods represent a bifurcation of the climate system and climate models have been shown to struggle in realistically simulating them (Calov and Ganopolski, 2005;Colleoni et al., 2014b). The glacial inception that has been studied most extensively is the one starting from the end of Last Interglacial (LIG, 125ka), corresponding to MIS 5a (Calov et al., 2005;Capron et al., 2017;Clark and Huybers, 2009;Crucifix and Loutre, 2002;Kubatzki et al., 2000;Nikolova et al., 2013;Otto-Bleisner et al., 2017;Pedersen et al., 2017). To our knowledge the penultimate interglacial, MIS 7 (240ka-170ka), has not received as much attention in ice-sheet modeling (Fig. 1). MIS 7 (Fig.1c and 1d) is the coldest interglacial occurring after the Mid Brunhes Event (MBE, ~430k) (Colleoni and Masina, 2014) with an intensity comparable to typical pre-MBE interglacials (Pages, 2016). Furthermore, $CO_2$ concentrations were lower than 260 ppmv for most of MIS 7. Contrary to the classic sawtooth pattern of the Earth's glacial cycle (gradual buildup and fast termination of ice sheets within 80,000-120,000 years) (Clark et al., 2009;Hays et al., 1976), the global ice volume during MIS 7e-7c increased rapidly and then decreased rapidly by around 60 m of global sea level equivalent (SLE, relative to present day) within a period of 20 ky (thousand years) (Cheng et al. (2016), Fig. 1). This is the

fastest such glaciation and deglaciation transition during the last 800 ky (Waelbroeck et al., 2002;Lisiecki and Raymo, 2005;Bintanja et al., 2005) although the last deglaciation had a sea level rise of ~100m in 10ky, which makes it an interesting test-case for coupled climate-ice-sheet models. Subsequently, the system stayed in a relatively stable interglacial state and descended into the next glacial state at the end of MIS 7a (~190ka) into MIS 6e. In this paper, we follow the lettering convention of MIS substages as suggested by Railsback et al. (2015). In summary, the climate system started from an interglacial and went into a glacial state for both MIS 7e-7d (235-225ka) and MIS 7a-6e (190-180ka) transitions, but bounced back to the interglacial state only for the MIS 7d-7c (225-215ka) state and not for MIS 6e-6d (180-170ka) (Fig. 1). The drivers of this unique phasing and amplitude of sea level high stands during MIS 7 still remain elusive. Both periods of pre-inception (MIS 7e-7d and 7a-6e) have similar orbital forcings and so do the periods post-inception (MIS 7d-7c and 6e-6d). But the $CO_2$ values differ by ~40ppmv. Although the $CO_2$ trends are similar over pre-inception, they differ in the post-inception times (Fig. 4a, Lüthi et al. (2008)). The $CO_2$ values rise steeply over MIS 7d-7c but stay low during MIS 6e-6d.

Most simulations struggle in realistically simulating the whole of MIS 7. Previous studies focusing on the MIS 7 have modeled MIS 7e and MIS 7a-7c as separate interglacials. For instance, Yin and Berger (2012) and Colleoni et al. (2014a) have simulated the climate during MIS 7e using LOVECLIM (Goosse et al., 2010) and CESM (Gent et al., 2011) respectively, and compared it to that during MIS 5. While Yin and Berger (2012) report the insolation-induced cooling during MIS 7e to be the primary reason for it being a cold interglacial, Colleoni et al. (2014a) suggest that 70% of the cooling over the Northern Hemisphere (NH) in MIS 7e compared to MIS 5 can be explained by $CO_2$ forcings. Further, Colleoni et al. (2014b) used CESM output to force an offline 3-D ice-sheet model (ISM), Grenoble Ice Shelf and Land Ice model (Ritz et al., 2001), and they were not able to produce as realistic results for MIS 7 as they could for MIS 5. Ganopolski and Brovkin (2017) simulated the last 400ky using the CLIMBER-2 EMIC (Petoukhov et al., 2000) coupled with the SICOPOLIS ISM (Greve, 1997). When forced with both orbital and $CO_2$ variations, they reported an exaggerated inception at MIS 6e (~180ka) followed by an overshoot to interglacial levels at MIS 6d (~160ka), while forcing with just orbital variations led to a much weaker glacial inception at MIS 6e. More recently, Willeit et al. (2019) performed transient simulations using the previous setup (CLIMBER-SICOPOLIS) forced with orbital, regolith removal and volcanic outgassing and also showed an overshoot to interglacial levels after the glacial inception of MIS 6e.

Our study presents transient simulations over the MIS 7-6 period which use a novel bidirectionally coupled 3-dimensional EMIC-ISM framework (LOVECLIM-PSUIM) with interactive ice sheets in both hemispheres. Using multiple ensemble runs, we test the sensitivity of the simulation to different forcing and model parameters. By comparing the MIS 7e-7c and MIS 7a-6d transitions, we investigate the relative role of orbital and $CO_2$ forcings on glacial inceptions and terminations. We also look at different climate-ice sheet feedbacks and local processes that induce a bifurcation in the system and can show abrupt changes in the climate-cryosphere system such as those in the Atlantic Meridional Overturning Circulation (AMOC).

In Sect. 2, the individual components of our coupled model and the coupling framework are described, along with a list of the experiments. Next, the main results are presented in Sect. 3, including multi-parameter ensemble simulations of ice sheet evolution, effects of orbital and $CO_2$ forcings pre and post glacial inception, abrupt changes in the climate-cryosphere system and the existence of multiple ice sheet equilibrium states. We conclude with a discussion of the key results and their implications for other glacial cycles along with key deficiencies in the current setup and possible solutions for future simulations.

## 2 Methods

We perform a series of transient glacial inception simulations covering the period from 240-170 ka using the bi-directionally coupled LOVECLIM-PSUIM system, henceforth called *LOVECLIP*. Both LOVECLIM (Friedrich et al., 2016;Nikolova et al., 2013;Timmermann and Friedrich, 2016;Timmermann et al., 2014;Yin and Berger, 2012) and the Penn State University Ice sheet Model, PSUIM (DeConto and Pollard, 2016;Gasson et al., 2018;Pollard et al., 2015;Tigchelaar et al., 2018), have been extensively used for simulating past and future climate. The individual components of the modelling framework as well as their coupling strategy are described below.

### 2.1 LOVECLIM

LOVECLIM is a three-dimensional Earth System Model of Intermediate Complexity with atmosphere, ocean, sea ice and vegetation models coupled together (Goosse et al., 2010). The atmospheric component of LOVECLIM, ECBilt (Opsteegh et al., 1998), is a spectral T21 ($5.625° \times 5.625°$) quasi-geostrophic model with three vertical levels including a parameterization of ageostrophic terms. The effect of $CO_2$ variations with respect to the reference $CO_2$ concentration (356 ppm) on the longwave radiation flux is scaled up by a factor $\alpha$ (Eq. 1), to account for the low default sensitivity of ECBilt to changes in $CO_2$ concentrations (Friedrich and Timmermann, 2020;Timmermann and Friedrich, 2016;Timm et al., 2010). The effect of $CO_2$ on the longwave radiation is given as:

$$LWR = \alpha . a(\lambda, \phi, p, t_{season}) . \log\left[\frac{CO_2(t)}{CO_2^{ref}}\right] \tag{1}$$

where $LWR$ is the longwave radiation flux from $CO_2$; $\alpha$ is our scaling factor for the transfer coefficient, $a$, which is a function of longitude, latitude, height and season; $CO_2^{ref}$ is the reference $CO_2$ value set at 356 ppm. $\alpha$ changes the sensitivity of our model. For reference, the equilibrium climate sensitivity for $CO_2$ doubling is 3.69K for $\alpha$ of 2. Climate sensitivity is a non-trivial measure that can be changed in many different ways. For instance, changing the cloud parameterization or surface parameters would change both the longwave and shortwave forcings. Adjusting multiple parameters may not necessarily lead to more realistic simulations. While it is possible that the climate sensitivity to some of the other forcings are also weak (Timm and Timmermann, 2007), we use a simple alpha ($\alpha$) parameter to change only the longwave sensitivity to $CO_2$, and not to other greenhouse gases. $\alpha$ is determined based on transient past and future simulations.

The ocean component, CLIO (Goosse and Fichefet, 1999), is a free-surface primitive-equation ocean general circulation model with $3° \times 3°$ horizontal resolution and 20 vertical levels; which is further coupled to a thermodynamic-dynamic sea ice model. Additionally, an iceberg model is employed that integrates iceberg trajectories (based on Coriolis force, air-water-sea ice drag, horizontal pressure gradient and wave radiation) and melt (depending on basal plus lateral melt and wave erosion along individual iceberg pathways) (Schloesser et al., 2019;Bigg et al., 1997). The atmosphere-ocean coupling is based on the freshwater, heat and momentum flux exchanges. The Bering Strait is opened and closed interactively depending on global mean sea level height. Specifically, its parameterized transport is multiplied with a constant that is zero for sea levels lower than – 50 m meters and linearly increases to 1 as global sea level rises to -25 m relative to present day. The terrestrial biosphere component of LOVECLIM, VECODE (Brovkin et al., 1997), estimates the evolution of vegetation cover (fraction of grass, trees and desert) over each land grid cell not covered by ice.

### 2.2 PSUIM

PSUIM is a hybrid ice-sheet-ice-shelf model that combines the scaled shallow ice and shallow shelf approximations (Pollard and DeConto, 2012b). It has been shown to reasonably capture both slow and fast flowing grounded ice regimes as well as floating ice shelves while being simpler and more computationally efficient than Full-Stokes or Higher-Order models. The

model also accounts for free grounding-line migration based on a sub-grid parameterization that calculates ice fluxes at the grounding line based on Schoof (2007). The ice is determined to be grounded or floating based on buoyancy, as per:

$$\left.\begin{array}{l} \rho_w(S - h_b) < \rho_i h; \text{ and } h_s = h + h_b; \text{ for grounded ice (ice sheet)} \\ \rho_w(S - h_b) > \rho_i h; \text{ and } h_s = S + h\left(1 - \frac{\rho_i}{\rho_w}\right); \text{ for floating ice (ice shelves)} \end{array}\right\} \tag{2}$$

where $\rho_w = 1028 kg m^{-3}$ is the density of ocean water; $\rho_i = 910 kg m^{-3}$ is the ice density; $S$ is the sea level ($m$); $h_b$ is the bedrock elevation ($m$); $h$ is the ice thickness ($m$); and $h_s$ is the ice surface elevation ($m$).

PSUIM calculates the surface energy and mass balances, by including the temperature and radiation contributions, to solve for surface melting and freezing (Robinson et al., 2010;Van Den Berg et al., 2008). Specifically, the energy flux available for melting ($dE > 0$) or refreezing ($dE < 0$) is given by:

$$dE = b(T - T_o) + (1 - a)Q - m, \tag{3}$$

where constant $b = 10\ Wm^{-2}K^{-1}$, $T$ is the surface air temperature, $T_o$ the freezing point, $a$ the albedo, $Q$ the surface incoming short wave radiation, and parameter $m$ a constant (see Sect.2.3). The albedo is linearly interpolated between values for no snow ($a_{ns} = 0.5$), wet snow ($a_{ws} = 0.6$), and dry snow ($a_{ds} = 0.8$),

$$a = (1 - r_s)a_{ns} + r_s[r_l a_{ws} + (1 - r_l)a_{ds}], \tag{4}$$

where $r_s$ is the snow covered area fraction and $r_l$ the ratio between liquid water contained in the snow mass and the maximum embedded liquid capacity. The parameter '$m$' in Eq. (3) represents net upwards infrared radiation from a solid surface at temperature $T_o$ to the atmosphere, plus a constant correction for other simplifications in Eq. (3) and has units of Wm$^{-2}$. The surface mass balance is then calculated based on snow fall (which is calculated locally based on total precipitation and temperature) and melting/refreezing based on $dE$. This surface mass balance is calculated at timesteps of 3 hours. Monthly surface air temperature ($T$) and surface incoming shortwave radiation ($Q$) (obtained from LOVECLIM in the current setup, discussed further in Sect. 2.3) are interpolated into sub-daily values in two steps. Firstly, the monthly values are interpolated to daily values using a weighted averaging of the values across two adjacent months. Next, a sinusoidal cycle with max temperature at 1400 and minimum at 0200, with a peak-to-peak amplitude of 10˚C, is superimposed on the daily data to account for diurnal variations.

The sub-ice-shelf ocean melting is calculated as per Pollard et al. (2015) using ocean temperature at 400m depth ($T_o$) from LOVECLIM as follows:

$$OM = \frac{K K_T \rho_w c_w}{\rho_i L_f}|T_{oc} - T_f|(T_{oc} - T_f) \tag{5}$$

where $OM$ is the subshelf ocean melting rate ($m yr^{-1}$), $T_{oc}$ is the LOVECLIM ocean temperature at 400m depth (℃), $T_f$ is the ocean freezing temperature at depth $z$ (m) calculated as $T_f = 0.0939 - 0.057 \times 34.5 - 0.000764 \times z$ , $K_T = 15.77 m yr^{-1}K^{-1}$ is a coefficient; $K$ is a non-dimensional factor of order 1, $K = 3$; $c_w = 4218 J kg^{-1}K^{-1}$ is the specific heat of ocean water; and $L_f = 0.335 \times 10^6 J kg^{-1}$ is the latent heat of fusion.

Calving is primarily parameterized depending on the ice shelf flow divergence. The model also includes parameterizations for surface meltwater and rainfall-driven hydrofracturing and the structural failure of tall sub-aerial ice cliffs, which produce strong ice retreat in Antarctic marine basins needed to explain past high sea level stands suggested by geologic data (DeConto and Pollard, 2016;Pollard et al., 2015). The sea level dependence is implemented by the formulation of boundary processes, such as calving, flotation of ice, grounding line dynamics and sub-grid pinning by bedrock bumps, which also affects the grounding line flux (Schoof, 2007). PSUIM is used to simulate ice sheets in both hemispheres. The bedrock deformation is

calculated by an ELRA (Elastic Lithosphere Relaxing Asthenosphere) model, assuming a bedrock density of 3370 kgm[-3] and

an isostatic asthenospheric relaxation time of 3000 years (Pollard and DeConto, 2012b). The basal sliding velocity is defined as in Pollard and DeConto (2012b) and depends on the basal sliding coefficient:

$$\widetilde{u_b} = C'|\tau_b|^{\mu-1}\widetilde{\tau_b} \,, \tag{6}$$

where $\widetilde{u_b}$ is the basal sliding velocity, $\widetilde{\tau_b}$ is the basal stress; $\mu$ is the basal sliding exponent (=2); $C'$ is the basal sliding coefficient which is a function of the basal homologous temperature:

$$C' = (1-r)C_{froz} + rC(x,y) \,, \tag{7}$$

with $r = max[0, min[1, (T_b + 3)/3]]$; where $T_b$ (°C) is the basal homologous temperature relative to the pressure melting point ($T_m = -0.000866h$, $h$ being the ice thickness in $m$); and $C_{froz} = 10^{-20}$m yr$^{-1}$Pa$^{-2}$ (which cannot be zero to avoid numerical inconsistencies but is small enough to allow essentially no sliding). For Antarctica, the sliding coefficient $C(x,y)$ is deduced from the inverse modelling approach of Pollard and DeConto (2012a). For the NH, a binary sliding coefficient map

is used with higher sliding over present-day oceans ($C(x,y) = 10^{-6}$m yr$^{-1}$Pa$^{-2}$ representing deformable sediments), and low sliding over present-day land ($C(x,y) = 10^{-10}$m yr$^{-1}$Pa$^{-2}$ representing non-deformable rock).The model was tested at two resolutions for each hemisphere; for the NH, a longitude-latitude grid is used at either $1 \times 0.5°$ or $0.5 \times 0.25°$, and for Antarctica, a polar stereographic grid is used at either $40 \times 40$ km or $20 \times 20$ km. No significant differences in the results using the two resolutions were noticed for either hemisphere. All the results presented in this paper use $1 \times 0.5°$ for the NH

and $40 \times 40$ km for Antarctica.

## 2.3 LOVECLIP

Figure 2 shows the coupling algorithm employed in the current setup to exchange information across LOVECLIM and PSUIM, between alternating climate model and ice sheet runs (chunks). LOVECLIM chunks of length $T_L$ alternate with PSUIM chunks of length $T_P$ ($\geq T_L$). Here we define the acceleration factor $N_A = T_P/T_L$. An earlier version of this coupling algorithm was

used by Heinemann et al. (2014) for a different ISM (Ice sheet model for Integrated Earth system Studies (Saito and Abe-Ouchi, 2004) that was active only in the NH and did not include ice shelf-dynamics. The coupling strategy has the advantage of using asynchronous coupling to speed up climate simulations at millennial to orbital timescales (Friedrich et al., 2016;Tigchelaar et al., 2018;Timm and Timmermann, 2007;Timmermann and Friedrich, 2016). The fidelity of using the acceleration factor depends on how quickly the variables of interest equilibrate to the slowly evolving external boundary

conditions. Preliminary experiments (not shown) with different acceleration factors suggest that the simulated ice sheet evolution is relatively insensitive to $N_A$ for $N_A \leq 5$. Therefore, $N_A = 5$ is used for the simulations presented in this paper, providing a good compromise between the objective to simulate realistic ice sheet evolution and computational efficiency.

PSUIM uses surface air temperature ($T$), precipitation ($P$), solar radiation ($Q$), and ocean temperature at 400m depth ($T_o$) as

inputs from LOVECLIM. These are downscaled using a bilinear interpolation approach. The surface temperature and precipitation outputs from LOVECLIM which are used for the PSUIM surface mass balance are bias-corrected in the coupler, following Pollard and DeConto (2012b), Heinemann et al. (2014) and Tigchelaar et al. (2018).

$$T(t) = T_{LC}(t) + T_{obs} - T_{LC,PD} \tag{8}$$
$$P(t) = P_{LC}(t) \times P_{obs}/P_{LC,PD} \tag{9}$$

where $T$ is monthly surface air temperature and $P$ is monthly precipitation forcing from LOVECLIM at timestep $t$. Subscripts '$LC$', '$obs$' and '$LC, PD$' refer to LOVECLIM chunk output, observed present day climatology, and LOVECLIM present day control run, respectively. The observed present day climatology is obtained from the European Centre for Medium-Range Weather Forecasts reanalysis dataset, ERA-40 (Uppala et al., 2005). These LOVECLIM biases are calculated for PD

simulations using an LGM bathymetry. We did compare the biases between using a PD or LGM bathymetry, and while there were regional differences, the large-scale structure was found to be similar (not shown). The annual mean of the monthly mean bias correction terms $T_{obs} - T_{LC,PD}$ and $P_{obs}/P_{LC,PD}$ are presented in Fig. S1. Temperature biases in LOVECLIM for boreal summer (JJA) and austral summer (DJF) are shown in Fig. S2 for reference, since summer temperatures are more crucial for ice sheet growth and decay. Furthermore, a lapse-rate correction of 8°C km$^{-1}$ is applied to account for differences between LOVECLIM orography and PSUIM topography for the interpolated temperature, $T(t)$, and precipitation is multiplied by a Clausius–Clapeyron factor of $2^{\frac{-\gamma.\Delta H}{10°C}}$, with $\gamma.\Delta H$ being the temperature lapse-rate correction, to account for the elevation desertification effect (DeConto and Pollard, 2016):

$$T_{PSUIM}(t) = T_{interp}(t) - \gamma.\Delta H \tag{10}$$

$$P_{PSUIM}(t) = P_{interp}(t) \times 2^{\frac{-\gamma.\Delta H}{10°C}} \tag{11}$$

where $T_{PSUIM}$ and $P_{PSUIM}$ are the final temperature and precipitation inputs for PSUIM, $T_{interp}$ and $P_{interp}$ are bias corrected LOVECLIM temperature ($T$, Eq. 8) and precipitation ($P$, Eq. 9) interpolated to PSUIM resolution, $\gamma$ is the lapse rate (8°C km$^{-1}$), and $\Delta H$ is the altitude difference between PSUIM grids and the corresponding LOVECLIM grid. Colleoni and Liakka (2020) used a similar fixed atmospheric lapse rate correction during downscaling temperature to their ice model, GRISLI, with $\gamma$ as 3.3°C km$^{-1}$ for annual mean and 4.1°C km$^{-1}$ for summer mean. And they reported slightly smaller ice sheets on using an elevation dependent lapse rate, going all the way up to 7.9°C km$^{-1}$. Instead of using a fixed value of $\gamma$, both Roche et al. (2014) and Bahadory and Tarasov (2018) used a dynamic lapse rate, where $\gamma$ is estimated locally for the ice model grids in each LOVECLIM grid. Moreover, the lapse rate also depends on the atmospheric $CO_2$ concentration. Such dynamic lapse rate corrections are not implemented in the current setup, and neither is the advective precipitation downscaling scheme of Bahadory and Tarasov (2018).

Surface incoming shortwave radiation from LOVECLIM ($Q$, Eq. (3)) is bilinearly interpolated to PSUIM grid and then used to calculate the surface mass balance. PSUIM calculates albedo using snow covered fraction ($r_s$) and the ratio between liquid water contained in the snow mass and the maximum embedded liquid capacity ($r_l$) from the last year of its previous chunk. This provides a more realistic estimate of the albedo than downscaling from LOVECLIM. $T_{oc}$, used in calculating the ocean sub-ice-shelf melting (Eq. 5) is interpolated to PSUIM grid using conservative remapping. For some of the floating ice (shelves) in PSUIM (Eq. 2), the ocean points underneath may not get an ocean temperature assigned on interpolation from LOVECLIM, since the land-sea mask in LOVECLIM is kept constant. For each of such grid points, the algorithm averages over the neighboring eight PSUIM grid points and this process is repeated until all PSUIM ocean points get ocean temperatures.

LOVECLIM orography and surface ice mask are updated based on the evolution of ice sheets and bedrock elevation from PSUIM. The PSUIM topography is upscaled to LOVECLIM grid using simple weighted averaging. Each grid cell of ECBilt and VECODE is defined as either ice-free or ice-covered (not fractionally covered); it is ice-covered if more than half of the cell has more than 10 m of ice in the finer PSUIM cells that lie within the LOVECLIM cell, thus changing the ground albedo to ice albedo. The total meltwater from basal melting and liquid runoff in PSUIM is dynamically routed based on PSUIM topography till it reaches the ocean or the domain edge, and then is routed to the nearest ocean grid point in LOVECLIM. The calving flux is channeled into CLIO's iceberg model (Schloesser et al., 2019;Jongma et al., 2009) in the Southern Hemisphere (SH) and as an iceberg melt flux (freshwater flux and heat flux) in the NH (Schloesser et al., 2019). While both freshwater flux and freshwater volume cannot be simultaneously conserved in an accelerated run (Heinemann et al., 2014), we conserve freshwater flux in the current setup. The primary rational for this being that surface freshwater and meltwater fluxes are balanced by the convergence of ocean salt fluxes in equilibrium states, and adjustments of the ocean circulation are thought to occur rapidly compared to those of ice sheets. PSUIM is forced by LOVECLIM precipitation and LOVECLIM by PSUIM

runoff. During a glacial inception (termination), the runoff from PSUIM into LOVECLIM reduces (increases) as ice sheets grow. This reduction (increase) in runoff into the ocean, relative to the evaporation, increases (decreases) the salinity of the ocean. This salinity change is spatially resolved.

The contributions of the ice sheets to global sea level changes are calculated independently for the two hemispheres in PSUIM, and the net sea level change is used for the next chunk of ice model run. Note, however, that LOVECLIM does not see the change in sea level, and the ocean bathymetry and land-sea mask (with the exception of the Bering Strait opening and closing) are not updated in the coupling framework. Our coupled simulations use the LGM bathymetry and land-sea mask throughout the entire transient simulation.


### 2.4 Experiments

The LOVECLIP experiments are initialized using present day ice sheet conditions and spun up using orbital and greenhouse gas (GHG) forcings of 240 ka for a period of 10ky. The model equilibrates to an ice sheet distribution in the NH corresponding to -20m SLE, implying an open Bering Strait. Our initial ice sheet distribution at 240ka is shown in Fig. 4c and is in close
agreement with that used by previous studies such as Colleoni and Liakka (2020) for 239ka and Colleoni et al. (2014b) for 236ka. From these initial conditions, LOVECLIP is run forward with two transient forcings: Orbitally induced solar insolation variations following Berger (1978); and time-varying atmospheric GHG concentrations measured from the European Project for Ice Coring in Antarctica Dome C ice core (Loulergue et al., 2008;Lüthi et al., 2008;Schilt et al., 2010). Two additional sets of experiments are run to discern the independent effects of the two primary forcings: (1) Time varying orbital forcing with
constant GHG concentration (set at its value for 240 ka), and (2) Constant orbital forcing (set at the orbit for 240 ka) and time varying GHG concentrations.

Furthermore, sensitivity experiments with different GHG sensitivities ($\alpha$, Sect. 2.1) and melt parameterizations ($m$, Sect. 2.2) are run with full forcing. Generally, higher $\alpha$ leads to a stronger sensitivity to $CO_2$ concentrations, and higher values of $m$
strengthen buildup and weaken melting of ice during interglacial climates. These experiments are presented in the first row of Table 1 (1-15) and Fig. 3. Additional simulations with different combinations of acceleration ($N_A$), GHG sensitivity ($\alpha$), melt parameter ($m$), basal sliding coefficient maps over the NH ($C(x, y)$) and higher ice model resolution ($0.5 \times 0.25°$ for NH, $20 \times 20$ km polar stereographic for Antarctica$)$ have been performed (experiments 16-50 in Table 1). The whole ensemble of simulations is presented in Fig. S3. Although we note that these experiments do not present a systematic evaluation of the full
parameter space, ice sheet trajectories are consistent with and thereby support the conclusions presented in this paper. For benchmarking purposes, our model throughputs ~1200 years per day on one node at the University of Southern California Center for High-Performance Computing, so for an acceleration ($N_A$) of 5, we simulate 6000 years per day in real time.

### 3 Results

### 3.1 Overview of multi-parameter ensemble coupled simulations

Figure 3 shows the simulated ice volume in SLE (m) from experiments that best describe parameter sensitivities, while the complete ensemble of simulations is presented in Fig. S3. Figure 3a shows the most realistic simulation (referred to as baseline simulation BLS, experiment 1 in Table 1) in comparison to the sea level reconstructions of Spratt and Lisiecki (2016). Parameter sensitivities will be further discussed below. The model captures the overall trajectory of ice volume evolution
reasonably well. Specifically, the model stays within the uncertainty range for the extreme glaciation-deglaciation event of

MIS 7e-7d-7c. Larger differences only exist as the glaciation into MIS 6 is delayed by ~3ky in the simulation (191ka instead of 194ka). A possible explanation for this discrepancy may be related to the temporal uncertainty in reconstructions themselves, since a similar lag occurs in other modeling studies (e.g., Ganopolski and Calov (2011); Ganopolski and Brovkin (2017). Higher climate sensitivity ($\alpha$) leads to a faster and stronger glacial inception and termination at the MIS 7e-7d-7c transition,

in response to the time-varying orbital forcing and $CO_2$ changes (Fig. 3b). Increasing the value of the melt parameter, $m$ (Eq. (3), leads to a deeper inception and much weaker termination (Fig. 3c). The most realistic simulation is obtained for $\alpha = 2$ and $m = 125$ Wm$^{-2}$. Unless otherwise mentioned, all results presented in this study are from this particular ensemble member (labelled BLS). When the climate sensitivity ($\alpha$) or the parameter in the linear energy balance model ($m$) exceed certain thresholds, our model simulates an unrealistic runaway glaciation. The model physics underlying this feature will be described

further below. Following the concerns raised in Edwards et al. (2019), we ran a BLS simulation with hydrofracturing and cliff instability inactive and found no differences in the ice evolutions. This finding illustrates the complexity of the task to better constrain associated parameters in comparison of paleo climate simulations and data.

## 3.2 Ice sheet evolution

The rapid waxing and waning of ice sheets during the MIS 7e-7d-7c transition is presented in terms of maps of ice height and basal velocities in Fig. 4. In our simulations, the primary contribution to the ice volume evolution during MIS 7-6 comes from the NH (blue line in Fig. 4b). SH only contributes around 10m SLE during the interstadial of MIS 7c (red line in Fig. 4b). During the short transition (~10ky) from MIS 7e (235ka, Fig. 4c) to MIS 7d (226ka, Fig. 4e), ~4km thick Laurentide and ~3km thick Cordilleran ice sheets are built up over the NH. The Eurasian ice sheet grows substantially slower in our simulations

during this period. Although the contribution of Antarctica during MIS 7d is less than 5m, the Filchner-Ronne ice shelf spreads further out into the Weddell Sea (Fig. 4f, ice shelves do not directly contribute to sea level change). This quick glaciation event coincides with decreasing NH summer insolation and $CO_2$ forcings. Although NH insolation reaches a minimum at ~230ka and starts rising again, $CO_2$ stays higher than 220ppm till ~227ka and drops only just before MIS 7d (~226ka, Fig. 4a). Subsequently, a rapid deglaciation event occurs, associated with a steep increase in both orbital and GHG forcings over MIS

7d to 7c. Our model successfully simulates ice sheet retreat similar to a saddle collapse of Laurentide and Cordilleran splitting (Gregoire et al., 2012). While some studies have suggested the sea level peak at MIS 7c to be lower (Dutton et al., 2009) than those at MIS 7e and 7a, our model simulates MIS 7c to be the highest peak in MIS 7 with marked deglaciation of the Laurentide and Cordilleran, reduced Innuitian, Greenland, and small Icelandic and Norwegian ice sheets (Fig. 4g), along with a reduced West Antarctic ice sheet (Fig. 4h). The highest contribution of SH to MIS 7 of ~10m (Fig. 4b and 4h) occurs during MIS 7c.

After a relatively stable interglacial state till MIS 7a, the system moves into a glacial state. At the end of the simulation, our model simulates a bigger Laurentide and relatively smaller and detached Cordilleran as the model glaciates into MIS 6 (Fig. 4i).

In the context of previous modelling studies and geological records over this MIS 7-6 period, our ice sheet distribution at MIS

7c (212ka, Fig. 4g and 219.5ka, Fig. S7) is very similar to that reported in Colleoni and Liakka (2020). However, we simulate a stronger inception compared to that of Colleoni et al. (2014b) over the corresponding 236-230ka period. They also reported a bifurcated but connected North American ice sheet at MIS 6 (157ka) from both their control (100km) and high resolution (40km) experiments. Our simulation results in separate Laurentide and Cordilleran ice sheets but generates neither a Eurasian nor a Siberian ice sheet, albeit at 170ka. On a side note, our North American ice sheet distribution at 180ka (Fig. 7) is closer

to that of Colleoni and Liakka (2020) at 157ka. Studies of NH reconstructions during MIS 6 such as Svendsen et al. (2004), over 160-140ka, Rohling et al. (2017), around 140ka, and Batchelor et al. (2019), over 190-132ka, have all reported glacial geological records to indicate a larger extent of the Eurasian ice sheet at MIS 6 glacial maximum compared to the LGM, while

our simulations only show a persistent Fenno-Scandian ice sheet and a relatively small Eurasian ice sheet at 170ka. More recently, Zhang et al. (2020) reported the existence of a Northeast Siberia-Beringian ice sheet at MIS 6e (190-180ka) using NorESM-PISM simulations validated by North Pacific geological records. However, our model does not simulate any ice over Alaska, Beringia and northeast Siberia over MIS 7-6.

Our model's difficulty in simulating the Eurasian ice sheet can be attributed to the competition between Laurentide and Eurasian ice sheet growth, which makes it arduous to realistically simulate them simultaneously alongside generating the right atmospheric patterns. Some previous studies have suggested that teleconnections from stationary wave patterns induced by a large Laurentide ice sheet could lead to warming over Europe and influence Eurasian ice sheet evolution (Roe and Lindzen, 2001;Ullman et al., 2014). The Laurentide building up first in our simulations could have changed the storm tracks and dried out Eurasia. It is also worth reiterating that LOVECLIM has a coarse T21 grid with a simple 3-layered atmosphere. While the circulation changes reported here maybe model dependent, Lofverstrom and Liakka (2018) reported that at least a T42 grid was needed in their atmospheric model (CAM3) to generate a Eurasian ice sheet using SICOPOLIS, albeit for the LGM. They attribute this discrepancy to lapse rate induced warming due to reduced and smoother topography and higher cloudiness leading to increased re-emitted longwave radiation towards the surface. These teleconnection patterns are further discussed in Sect. 3.6. Our LOVECLIM setup also uses a fixed lapse rate for downscaling LOVECLIM surface temperatures (Eq. 10 and 11), while both Roche et al. (2014) and Bahadory and Tarasov (2018) used a dynamic lapse rate, which is estimated locally for the ice model grids in each LOVECLIM grid. Bahadory and Tarasov (2018) reported ice thickness differences up to 1km on using the dynamic lapse rate scheme compared to a fixed $6.5°Ckm^{-1}$. Nevertheless, for runaway trajectories, our model can build up a Eurasian ice sheet for ice volumes greater than -200m SLE once the Laurentide growth slows down (not shown). Our modelling setup also does not account for sub-grid mass balances, which can be especially relevant over mountainous regions with large sub-grid relief such as Alaska (Le Morzadec et al., 2015). Coarse grids tend to average out tall peaks and low valleys and thus don't capture the non-linear combination of accumulation zones on the high peaks and ablation zones in the valleys. These shortcomings could explain the lack of Eurasian, Siberian and Beringian ice sheets in our simulations.

## 3.3 Effects of orbital and GHG forcings

Figure 5 shows the effects of the individual orbital and GHG forcings on the simulated ice volume. As expected, keeping both orbital and GHG forcings fixed at 240ka values leads to no change in the ice volume (control run, dashed line). Forcing with only GHG variations alone (red line) leads to a small cooling trend compared to the control run but does not simulate any glacial inceptions. On the other hand, forcing with orbital variations only (blue line) does simulate glacial inceptions, albeit only half of the magnitude over MIS 7e-7d-7c, and the system does not glaciate completely at MIS 6 (170ka). This can be attributed to the fact that the NH summer insolation at 170ka is relatively strong at almost interglacial levels (Fig. 1a and 4a) and that the MIS 6 inception might have been controlled by the low GHG values. We also performed orbital only runs with different background $CO_2$ values of 180,200,220,240,260 and 280 ppmv, instead of keeping $CO_2$ constant at 240ka values (~245 ppmv); and found the model to still simulate the inception over MIS 7e-7d (not shown). Our orbital-only simulation is also very similar to the run of Ganopolski and Brovkin (2017) which was forced with orbital variations only with a constant $CO_2$ concentration of 240ppm (ONE_240 experiment, green line in their Fig. 8). Although they simulated the 7e-7d-7c transition well, their orbital-only run did not glaciate successfully into MIS 6. This suggests that glacial inceptions, at least over the MIS 7-6 period, are primarily controlled by orbital forcings, discussed further in the next Sect. 3.4, supporting previous studies of Ganopolski and Brovkin (2017), Yin and Berger (2012) and Ganopolski and Calov (2011). However, it is imperative to restate that orbital forcings alone cannot force the system into the MIS 6 glacial, and low GHG values over 180-170ka are crucial for the MIS 6 inception. Please note that all four runs in Fig. 5 were conducted with GHG sensitivity ($\alpha$) = 2 and melt

parameter ($m$) = 125 Wm$^{-2}$, and different values of the parameters might have led to different ice sheet evolutions and different sensitivities with respect to orbital and greenhouse gas forcing.

## 3.4 Effects of forcings pre and post inception

In Figures 4a and 4b we highlight two 20ky periods in shading (235-215ka and 190-170ka) in dark and light grey colors. The dark grey periods (235-225ka and 190-180ka) are characterized by minimum values of NH summer insolation and the buildup of ice volume. The light grey marked periods (225-215ka and 180-170ka) correspond to peak summer insolation. In case of MIS 7d-7c, we observe a glacial termination, whereas the MIS 6e-6d period is characterized only by small changes in ice volume. By compositing the climate evolution over the NH for these two periods, we can further explore the reasons for the varying ice sheet responses during the MIS 7d-7c and the MIS 6e-6d periods (Figure 6). MIS 7e-7d-7c (MIS 7a-6e-6d) data are marked by dashed (solid) lines. Figure 6a shows the similarity of the orbital forcings during both periods. In contrast, their respective $CO_2$ evolutions are very different (Figure 6b). Even though the $CO_2$ values are markedly different in the first part (10ky) of the two periods (Fig. 6b), the simulated total NH ice volume evolution is quite similar (Fig. 6c). This highlights the relevance of orbital forcing during glacial inceptions. In spite of orbital forcing being similar over the second half of the composite figure, model trajectories diverge markedly. The successful glacial termination coincides with increasing $CO_2$ concentrations (dashed line), whereas the aborted termination during MIS 6e-6d (180-170ka, solid line) is associated with flat-lined glacial $CO_2$ values (Fig. 6a-6c). Although the insolation maximum is weaker for the later period (MIS 7a-6e-6d) compared to the earlier period (MIS 7e-7d-7c), we argue the full inception of the later period to be primarily driven by changes in $CO_2$. This is further supported by the orbital-only run showing a substantially weaker inception and subsequent termination over MIS 7a-6e-6d compared to MIS 7e-7d-7c (Figure 5), suggesting that the latter full inception could not be attributed only to an insolation threshold. Likewise, the global average surface temperature evolution is similar for the pre-inception case but different for the post-inception cases (Fig. 6d).

For both periods the Atlantic Meridional Overturning Circulation (AMOC) is strongest during the inception phase, and weaker during termination (Fig. 6e). The AMOC weakening is substantially more pronounced in the earlier period, following the successful termination. In both periods, the AMOC recovers almost to its full interglacial state. The reduced AMOC (Fig. 6e) could lead to increased subsurface warming (Liu et al., 2009;Clark et al., 2020) causing the higher subsurface melting in Fig. 6i. Figure 6 also shows the evolution of the different mass balance terms for both the pre-inception and post-inception cases. Accumulation and ablation depend on the surface temperature over and the extent of the ice sheets (Fig. 6f and 6g). As ice sheets grow further equatorward, they come in contact with warmer moist air leading to a positive feedback on the ice growth because of higher moisture carrying ability of warm air. But higher temperatures also lead to higher ablation because of increased surface melting. Furthermore, as the ice sheet grows in height, accumulation decreases because of the elevation desertification effect (DeConto and Pollard, 2016), while ablation reduces due to the lapse rate. Although accumulation and ablation can change both in and out of phase, the delicate interplay of leads and lags between them governs the sign of the net surface mass balance (Fig. 6h). For the pre-inception cases, accumulation leads ablation producing a net positive surface mass balance (SMB) till it reaches peak glaciation for both time periods. Subsequently, the SMB turns negative only for the second half of the 7e-7d-7c period and not the 7a-6e-6d period (Fig. 6e-g). The deglaciation is initiated by increased ablation around 4ky after the inception at MIS 7d (~221ka, dashed line in Fig. 6g), corresponding to the increasing $CO_2$ (dashed line in Fig. 6b), followed by a decrease in accumulation (dashed line in Fig. 6f) that can be attributed to the ice sheets retreating further north. Also, the ablation over 7d-7c (225-215ka, Fig. 6g) contains an additional saddle collapse (Gregoire et al., 2012) contribution when the Laurentide and Cordilleran ice sheets separate (Fig. 4e) leading to higher surface melting followed by rapid melting of both ice sheets. This shows up as the sharp spike in ablation around 15ky of the 7e-7d-7c transition (~220ka,

Fig. 6g) and amounts to the steep negative SMB in Fig. 6h. Together with the negative subshelf melting spike in Fig. 6i, this leads to a ~10m rapid increase in SLE around 219.5ka in Fig. 6c and Fig. 4b. Such abrupt changes are discussed further in Sect. 3.5. Although the Laurentide and Cordilleran ice sheets separate during the period 180-170ka, the net SMB does not exhibit a negative trend. This can be attributed to the low $CO_2$ value (<200ppmv) leading to lower temperatures (Fig. 6d) and reduced ablation even if the Laurentide extends equatorward (Fig. 6g). Furthermore, the southern extent of the Laurentide can lead to changes in circulation patterns that can alter the SMB (discussed in Sect. 3.6).

**3.5 Abrupt changes in the coupled climate-cryosphere system**

One advantage of using a fully coupled framework is that feedbacks in the climate-ice-sheet system can be simulated and understood. Here we focus on the feedbacks that lead to exceptionally fast ice loss around 220ka during the MIS 7d-7e transition; with anomalies of different climate variables during this transition for 220.5ka, 220ka and 219.5ka shown in Fig. S4, S5 and S7 respectively. A 3˚C anomalous subsurface warming over Baffin Bay at 220.5k (Fig. S4b) causes the Laurentide ice-sheet to melt from the east (Fig. S4a, S4d and S4f). At the same time, as the very western margin of the Laurentide starts thinning, it becomes a floating ice shelf instead of being grounded, as can be seen by the grounding line and ice velocities in Fig. S4a. This is because as the ice-sheet retreats, land areas below sea-level become exposed, which are connected to ocean points. PSUIM then assumes that the points will become ocean points and therefore the thinning western Laurentide changes from a grounded ice sheet to an ice shelf (grounding line in Fig. S4a). This shelf on the western margin has a surface temperature relatively warmer than the rest of Laurentide (Fig. S4c) and shows a weakly positive to negative SMB anomaly (Fig. S4d and S4f). The surface melting in the ice free region between Laurentide and Cordilleran ice sheets leads to an expanded surface ablation zone (Fig. S4d) and accelerated mass balance-elevation feedbacks (Weertman, 1961). This sort of accelerated melting due to the saddle effect has previously been documented by Gregoire et al. (2012) for the Meltwater Pulse events. These anomalies over the western Laurentide amplify over the next 500 years. Alongside warmer temperatures over the eastern Laurentide, the western Laurentide also shows anomalies up to +2˚C during 220ka (Fig. S5c). Not only the floating shelves, but also grounded regions of the western Laurentide show negative SMB anomalies (Fig. S5d) and basal sliding (Fig. S5a). Further, relatively warm subsurface ocean water (>-1˚C) seeps along the west bank of Hudson Bay leading to a more pronounced negative mass balance (Fig. S5d). This shows up as a spike in the subsurface ocean melt values in Fig. 6i. Temporal snapshots every 0.1ky in the vicinity of the spikes in ablation (Fig. 6g), SMB (Fig. 6h) and ocean melting (Fig. 6i) are shown in Figure S6. It shows that the spikes in ablation and SMB predominantly come from a small area in the southern end of the Laurentide, while the spike in subshelf melting results only from the western part of the Laurentide with a receding grounding line. Although our model simulates sub shelf melting along the western Hudson Bay, we did not find any geologic evidence of such subsurface melting around 219.5ka. It is also worth mentioning that our setup does not simulate forebulges or other specific mechanisms modelled by more comprehensive full-Earth models. But Tigchelaar et al. (2018) have reported such changes in mass balance arising due to changing of ice sheets to ice shelves near the grounding line. The spike in subshelf melting (Fig. 6i) as well as surface ablation (Fig. 6g) during this period lead to an increase of the freshwater flux into the ocean. This could explain the synchronization with the AMOC slowdown seen during this period in Fig. 6e. However, since our model is run at an acceleration ($N_A = 5$) and we conserve the freshwater flux (Sec. 2.3), the total freshwater volume dumped into the ocean is being underestimated, which may distort the LOVECLIM response. These surface and sub-surface melting processes of the Laurentide trigger a rapid retreat of the ice-sheet within the next 0.5ky (Fig. S7), accounting for the ~10m SLE rise in 0.5ky (Fig. 6c and 4b). Such relatively ice-free conditions at 219.5ka (Fig. S7) were also reported by Colleoni and Liakka (2020) in both their control (100km) and high resolution (40km) simulations.

## 3.6 Climate-ice sheet bifurcations and multiple equilibria

Figure 3 indicates a strong sensitivity of the simulated ice-sheet evolution to the melt parameter ($m$). Experiments with values in the range of $120\,Wm^2 \leq m \leq 130\,Wm^2$ and other parameters as in BLS produce a realistic glacial inception and termination over MIS 7e-7d-7c. Lower $m$ values lead to a reduced magnitude of glaciation, while higher values cause a rapid glacial build-up and a run-away effect with unrealistic, unabated growth of ice-sheets. Even though we did not run steady-state experiments, this behavior is reminiscent of a saddle node bifurcation. Bifurcations in the climate-cryosphere system in response to astronomical forcings have been previously documented by studies such as Paillard (1998), Calov et al. (2005), Ashwin and Ditlevsen (2015) and Ganopolski et al. (2016). While previous studies have used empirical models or coupled ice sheet models to understand such bifurcations based solely on forcing and ice volume thresholds, here we investigate the changes in climate teleconnections and stationary wave patterns that can arise from slightly different ice sheet distributions, to explain the inherent mechanisms of the simulated bifurcation. Roe and Lindzen (2001) suggested that the topography of an ice sheet such as the Laurentide induces a high-pressure anticyclonic circulation over the western end of the ice sheet. The associated cooling and upslope flow lead to enhanced rainfall over the western and southwestern ends of Laurentide. However, the prevailing cold northerlies downslope cause a reduction in rainfall over the ice sheet. This interplay between cooling associated with anticyclonic circulations alongside enhanced rainfall over western Laurentide and the reduction in rainfall over most of the ice sheet due to cold northerlies, can lead to an equilibrium ice sheet configuration or ice sheet growth/decay. Figure 7 shows the ice volume evolutions and anomalies in climate and ice sheet variables at 180ka with respect to 240ka. The initial values of these variables at 240ka is shown in Fig. S8.

Figure 7a shows two simulations that have a very similar evolution and capture the 7e-7d-7c transition realistically but show very different trajectories after the inception at MIS 6. Both ensemble members were run with the same GHG sensitivity, $\alpha$=2, but different melt parameters, $m$=125 Wm$^{-2}$ and $m$=130 Wm$^{-2}$. While one leads to a stable inception into MIS 6 (blue, $m$=125 Wm$^{-2}$), the other leads to a runaway glaciation (black, $m$=130 Wm$^{-2}$) with a total ice volume of 180 m SLE at 160ka (Fig. 7a). We assume that the bifurcation of the trajectories happens around 180ka, where the difference in ice volume between the two ensemble runs is only 10m SLE. Although the Cordilleran looks very similar, the Laurentide is slightly bigger and has a higher and wider dome at the southern tip in the runaway simulation (Fig. 7c compared to 7b). Also, the Laurentide and Cordilleran are connected further south and this bridge is also higher in the runaway simulation (Fig. 7e). This higher bridge and Laurentide ice sheet locally lead to a surface cooling and thus to a net positive surface mass balance (Fig. 7f and 7g). Also, an anomalous cyclonic circulation develops south of the Laurentide in the runaway case leading to a positive net budget just south of the Laurentide (Fig. 7g), while the positive budget is much further away from the Laurentide in the stable run (Fig. 7f). Westerly storm tracks veered further south by a weaker Aleutian Low and a stronger North Pacific High, as reported by Oster et al. (2015) for the LGM, bring in moisture towards the southern end of the Laurentide. The shape of the Laurentide and the saddle in the stable run cause this jet stream to meander around and precipitate on the southeastern tip of the Laurentide (Francis and Vavrus, 2012) (Fig. 7h). These winds might also cause the moisture laden air from the Gulf of Mexico to precipitate just north of the Gulf of Mexico (Fig. 7h). But in the runaway run, the jet stream is more northerly and precipitates over the southwestern tip of Laurentide (Fig. 7i). The Laurentide ice-sheet extending further south causes the moist air from the Gulf of Mexico to also precipitate over the southwestern end of the Laurentide alongside the moist air from the jet stream intensifying further southwestward growth of the Laurentide. This southwestward growth of Laurentide in turn enhances the poleward moisture transport. While these stationary wave feedbacks are similar to the ones described by Roe and Lindzen (2001) and they suggested these patterns to be robust for a range of parameters, it is possible that such circulation patterns could change when using a more realistic atmospheric model or by the presence of a Eurasian ice sheet. The atmospheric patterns strengthen over the next 10ky and the runaway run simulates ~30m SLE greater ice volume than the stable run (Fig. S9). The difference between the two runs at 180ka and 170ka are presented in Fig. S10. As mentioned earlier in Sect. 3.2, it is important to

acknowledge the low horizontal and vertical resolutions of LOVECLIM's atmosphere, which could mean the circulation changes reported here to be model dependent.

## 4 Summary and Discussion

Modeling glacial cycles remains a key challenge, especially because of the two-way interactions between ice sheets and climate and the emerging possibility for multiple equilibrium states. The ice-volume evolution originates from a time-integration of small net mass balance terms (e.g. Fig. 6h), which themselves originate from the difference of large accumulation and ablation terms. Long-term orbital-scale integrations of such delicate net surface mass balances can further lead to an accrual of errors. Here we focused on the penultimate interglacial, MIS 7-MIS 6 which is characterized by intervals with both in-phase and out-

of-phase orbital and GHG variations. This interesting period involves one of the fastest glacial build-ups and terminations, with SLE variations of up to ±60m within a period of 20ky. Due to the rapid response of the ice sheets to orbital and $CO_2$ forcings, this period serves as an excellent benchmark for coupled climate-ice sheet simulations. Our bidirectionally coupled three-dimensional climate-ice sheet model simulations with ice sheets and ice shelves represented in both hemispheres suggest that glacial inceptions are more sensitive to orbital variations, whereas terminations from deep glacial conditions need both

orbital and $CO_2$ forcing to work in tandem over a narrow ablation zone at the southern margins of northern hemispheric ice sheets. We find that small changes in the Laurentide's ice distribution for similar total ice volumes reminiscent of a saddle node bifurcation, which in turn determines whether the coupled trajectory will follow a deglaciation or a runaway glaciation pathway in response to the combination of forcings. This runaway glaciation can be explained in terms of a positive stationary-wave-ice sheet feedback in which ice topography-driven moisture transport from westerly storm tracks, a cold high-pressure

anticyclonic circulation and moisture-laden winds from the Gulf of Mexico lead to enhanced rainfall accumulation over the southern tip of the Laurentide, making it grow further southwestward.

The simulated ice sheet volume is well within the range of reconstructions for a rather narrow range of parameters. Small changes in parameter values can produce strongly diverging trajectories, and the emergence of multiple equilibrium states may

also suggest the model's dependence on initial conditions. This poses a challenge, as many ice sheet and climate model parameters remain poorly constrained. In this context, we note that parameterizations associated with hydrofracturing and cliff instability did not impact our ice sheet trajectories. These processes have provided substantial contributions to the rapid Antarctic ice sheet retreat simulated in response to future climate projections (DeConto and Pollard, 2016), and better constraining these parameterizations is important to reduce uncertainties related to future sea level trajectories (e.g., Edwards

et al., 2019). Presumably, these processes did not play an important role in our present simulations, because the climate is generally too cold, suggesting that opportunities for constraining these parameters in glacial simulations may be limited. We further note that the parameter sets which allowed for the most realistic simulation of glacial inceptions during MIS 7-MIS 6 may not necessarily be optimal for other periods. That optimal parameter sets can depend on the period over which they are optimized, has recently been shown for a similar coupled climate ice sheet model (Bahadory et al., 2020).


Our present setup has difficulties in realistically simulating both Laurentide and Eurasian ice sheets simultaneously and generates a smaller Eurasian ice sheet compared to reconstructions, which could be a model dependent feature of LOVECLIM, given it is a T21 grid with only three levels in the atmosphere, and so could vary with the choice of the climate model used. Since we use an accelerated setup, we only conserve the freshwater flux from the ice model to LOVECLIM, which could lead

to an underestimation of the oceanic circulation changes due to the lesser volume of net freshwater being dumped into the ocean. Nevertheless, there is scope of further improving the current setup. For instance, we only implement temperature and precipitation bias corrections in the current setup, and including bias corrections for radiation and ocean temperature might

improve our representation of ice sheets. Future research might further improve the current setup by including the advective precipitation downscaling scheme (Bahadory and Tarasov, 2018) to account for orographic forcing, which is not captured in LOVECLIM. We are also investigating the possibilities of using a dynamical, an altitude-dependent and a $CO_2$-dependent lapse rate corrections while downscaling temperature from LOVECLIM to PSUIM. This is because the atmospheric lapse rate depends on the atmospheric $CO_2$ concentration – an effect that has not been considered so far in glacial dynamics. Furthermore, improving our basal sliding coefficient map for the NH using information of sediment sizes, instead of simply using a binary coefficient map, has the potential of further improving the simulations.

Potentially more realistic results could be obtained if the simulations were unaccelerated (which would be computationally very expensive), and from using more complex climate models that include stratification-dependent mixing in the ocean for instance. Furthermore, Glacial Isostatic Adjustment (GIA) processes captured only in comprehensive full-Earth models such as forebulges are not simulated in the ice-sheet model used here. Nevertheless, we would like to reiterate that simulating a trajectory is more difficult than conducting timeslice experiments, as climate and ice sheet components work on totally different timescales and a fine interplay of parameters can add up to very different equilibrium states. And such coupled climate-ice sheet paleo-simulations offer great opportunities for constraining parameter sets for future simulations.

**Data Availability**

The data that support the findings of this study are available from the corresponding author on request.

**Author contributions**

AT and DC designed the research. DC, FS, MH and DP developed the model code. DC conducted the model simulations and analyzed the data. DC and AT prepared the manuscript with contributions from all the co-authors.

**Competing interests**

The authors declare no competing interests.

**Acknowledgements**

This research is supported by the Institute for Basic Science, South Korea (Grant No: IBS-R028-D1) and NSF Grant # 1903197. The simulations were conducted at the Center for High-Performance Computing at the University of Southern California. We would like to thank Lev Tarasov and another anonymous reviewer for their constructive comments, which have definitely helped improve the quality of the manuscript. DC would also like to acknowledge the many helpful discussions at the Advanced Climate Dynamics Course (ACDC) summer school in 2018 and the ACDC 10-year alumni conference in 2019.

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

| Expt Number | Orb Forced | GHG Forced | $N_A$ | $\alpha$ | $m$ (Wm$^{-2}$) | $C$ (myr$^{-1}$Pa$^{-2}$)-NH |
|---|---|---|---|---|---|---|
| 1 (BLS) | **Y** | **Y** | **5** | **2** | **125** | |
| 2 | **N** | **N** | 5 | 2 | 125 | |
| 3 | Y | **N** | 5 | 2 | 125 | Binary distribution |
| 4 | **N** | Y | 5 | 2 | 125 | |
| 5 | Y | Y | 5 | 2 | 125 | **1.**Ocean: |
| 6 | Y | Y | 5 | **1.8** | 125 | $C(x,y)=10^{-6}$; |
| 7 | Y | Y | 5 | **2.2** | 125 | representing |
| 8 | Y | Y | 5 | **2.5** | 125 | deformable sediments |
| 9 | Y | Y | 5 | **3** | 125 | **2.**Land: |
| 10 | Y | Y | 5 | 2 | **80** | $C(x,y)=10^{-10}$; |
| 11 | Y | Y | 5 | 2 | **100** | representing non- |
| 12 | Y | Y | 5 | 2 | **120** | deformable rock. |
| 13 | Y | Y | 5 | 2 | **130** | |
| 14 | Y | Y | 5 | 2 | **140** | |
| 15 | Y | Y | 5 | 2 | **150** | |
| *16-20* | *Y* | *Y* | *5* | *1.5* | *120,125,130,140,150* | |
| *21-24* | *Y* | *Y* | *5* | *3.5* | *80,100,120,125* | |
| *25-27* | *Y* | *Y* | *1 (30ky run)* | *2* | *110,120,130* | |
| *28-30* | *Y* | *Y* | *2 (30ky run)* | *2* | *110,120,130* | *Binary* |
| *31-33* | *Y* | *Y* | *10* | *2* | *110,130,150* | |
| *34-36* | *Y* | *Y* | *10* | *2.5* | *110,120,130* | |
| *37-38* | *Y* | *Y* | *20* | *2.5, 3* | *125* | |
| | | | | | | *Tertiary* |
| | | | | | | **1.***Ocean:* |
| | | | | | | $C(x,y)=10^{-6}$; |
| | | | | | | **2.1** *Land (soft tills):* |
| *39-41* | *Y* | *Y* | *5* | *2* | *125* | $C(x,y)=\mathbf{10^{-7},10^{-8},10^{-9}}$ |
| *42-44* | *Y* | *Y* | *5* | *2* | *150* | *over northeastern* |
| *45-47* | *Y* | *Y* | *5* | *2.5* | *125* | *North America* |
| | | | | | | **2.2** *Land (hard bed):* |
| | | | | | | $C(x,y)=10^{-10}$ |
| *High Resolution Runs:* 0.5 × 0.25° *for NH*, 20 × 20 km *polar stereographic for Antarctica* | | | | | | |
| *48-50* | *Y* | *Y* | *5* | *2* | *110,130,150* | *Binary* |

**Table 1: List of all ensemble runs performed for the study (shown in Fig. S3). The first 15 experiments are discussed in Sect. 3.1 and shown in Fig. 3. Values in bold represent the difference from the baseline simulation (BLS, experiment number 1). $N_A$ represents the PSUIM vs LOVECLIM acceleration factor (Sect 2.3). α represents the GHG sensitivity scaling factor (Eq. 1, Sect. 2.1) and $m$ represents the constant parameter in the surface energy balance equation (Eq. 3, Sect. 2.2). $C$ represents the basal sliding coefficient map used for the NH (Eq. 7, Sect. 2.2).**


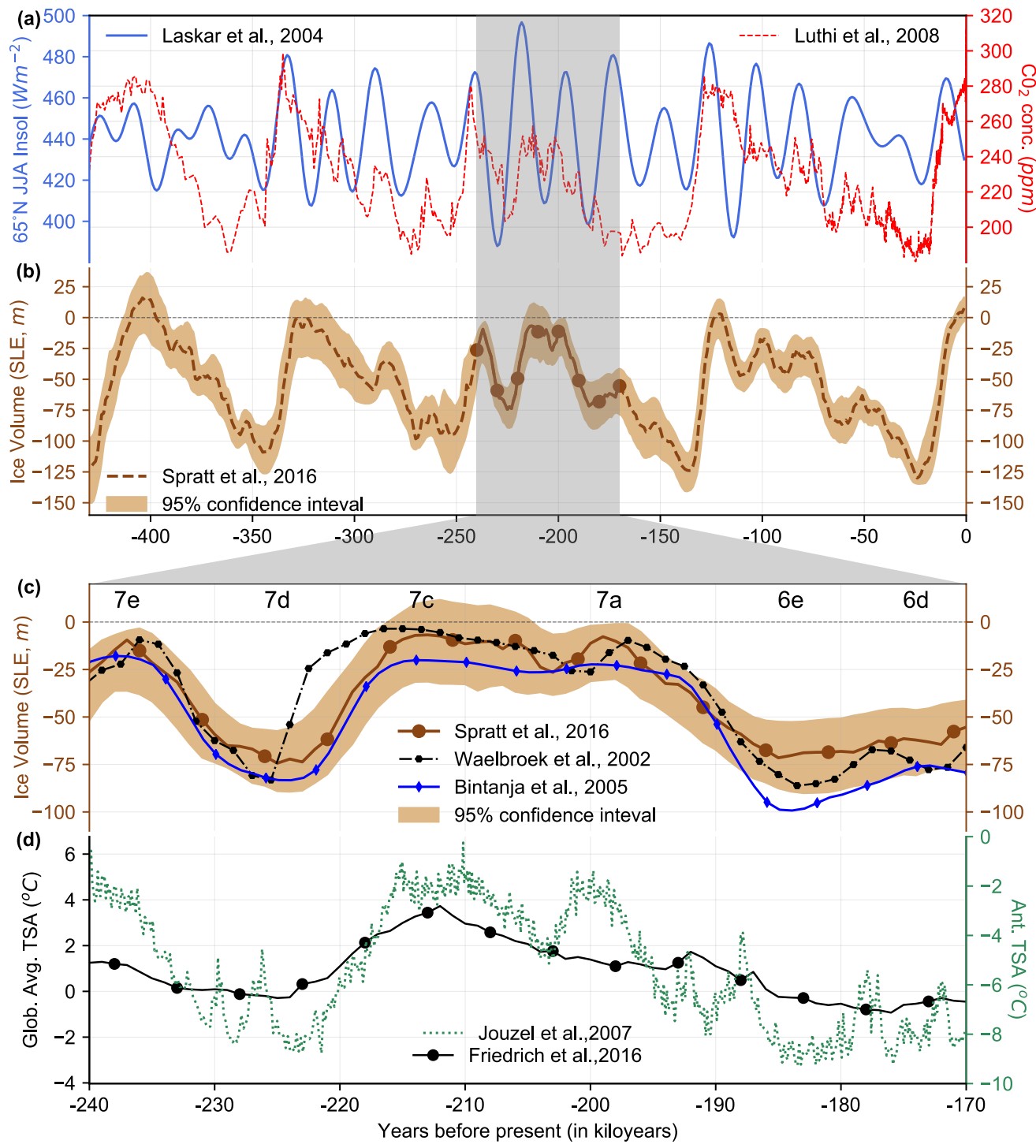


**Figure 1: Overview of the forcings and reconstructions relevant to this study. From top to bottom: (a) summer insolation at 65˚N (Wm⁻², blue (Laskar et al., 2004)) and CO₂ concentration (ppm, red (Lüthi et al., 2008)) over the last 430ka; (b) Sea level reconstructions (m) along with 95% confidence limits from Spratt and Lisiecki (2016) (brown) since the Mid Brunhes event. Notice**
**the relatively cold MIS 7; (c) Sea level reconstructions (m) from Spratt and Lisiecki (2016) (brown), Waelbroeck et al. (2002) (black) and Bintanja et al. (2005) (blue) over MIS 7 (240-170ka); (d) Global average surface air temperature anomaly reconstructed from proxies (˚C, black (Friedrich et al., 2016)) and Antarctic temperature anomaly relative to present day (˚C, green (Jouzel et al., 2007)). The lettering convention of MIS substages as suggested by Railsback et al. (2015).**

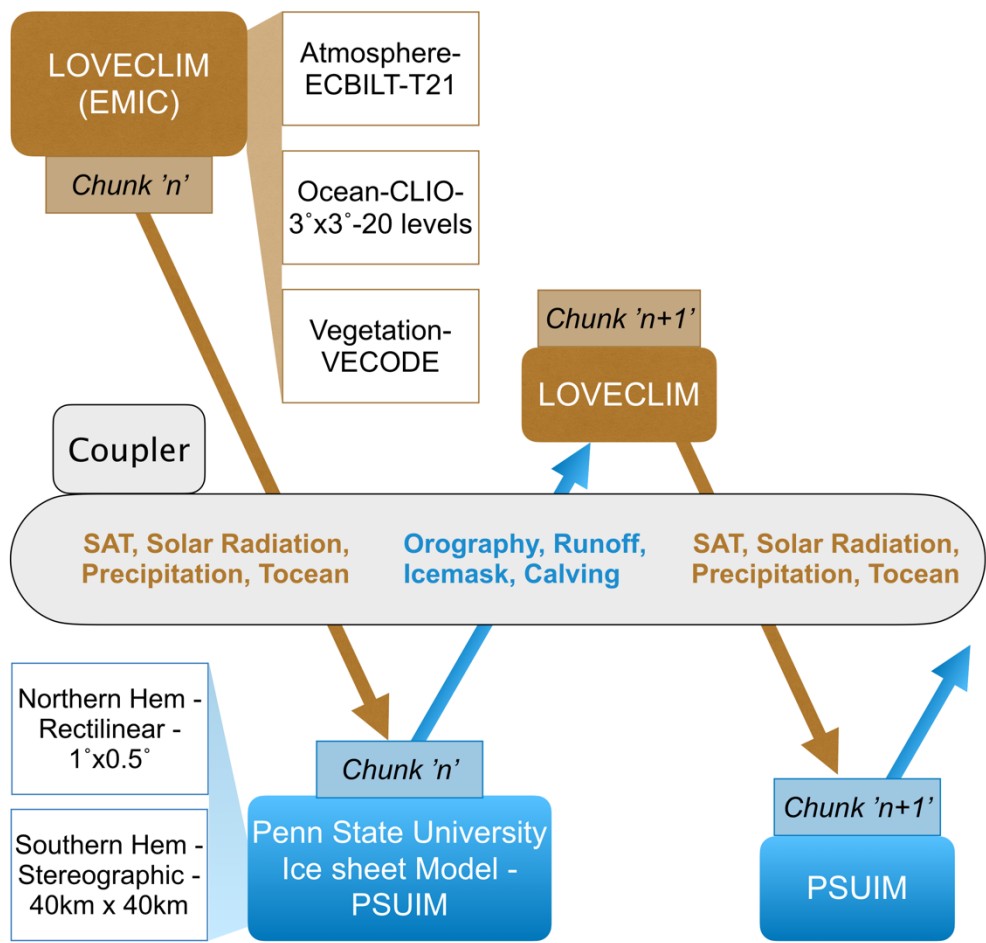


**Figure 2: Schematic of the coupling between LOVECLIM and PSUIM. SAT is the surface air temperature and Tocean is the ocean temperature at a depth of 400m. Refer to Sect. 2.3 for details.**

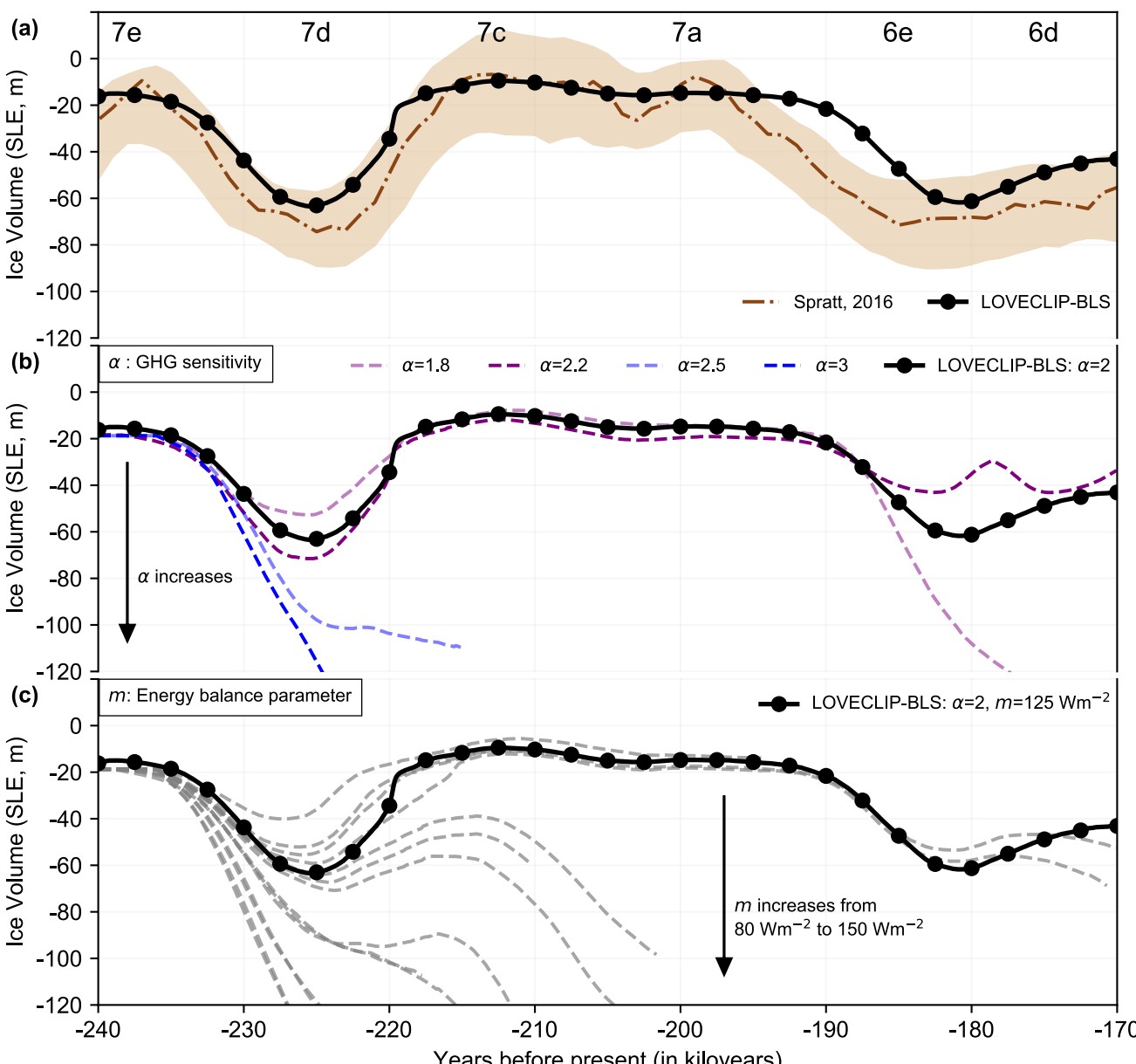

Figure 3: Transient simulation and parameter sensitivity over MIS 7. (a) Sea level reconstruction (m) of Spratt and Lisiecki (2016) (brown) and total ice volume (in terms of SLE, m) from LOVECLIP baseline simulation (BLS, Experiment 1 in Table 1 using $\alpha$=2 and $m$=125 Wm$^{-2}$). (b) LOVECLIP ensemble runs with varying GHG sensitivities ($\alpha$, Eq. (1)) and a melt parameter value ($m$, Eq. (3)) of 125 Wm$^{-2}$. The best results are obtained for an $\alpha$ of 2. (c) LOVECLIP ensemble runs with $\alpha$ of 2 and different values of the melt parameter ($m$, Wm$^{-2}$). The best results are obtained for an $m$ of 125 Wm$^{-2}$ (BLS).

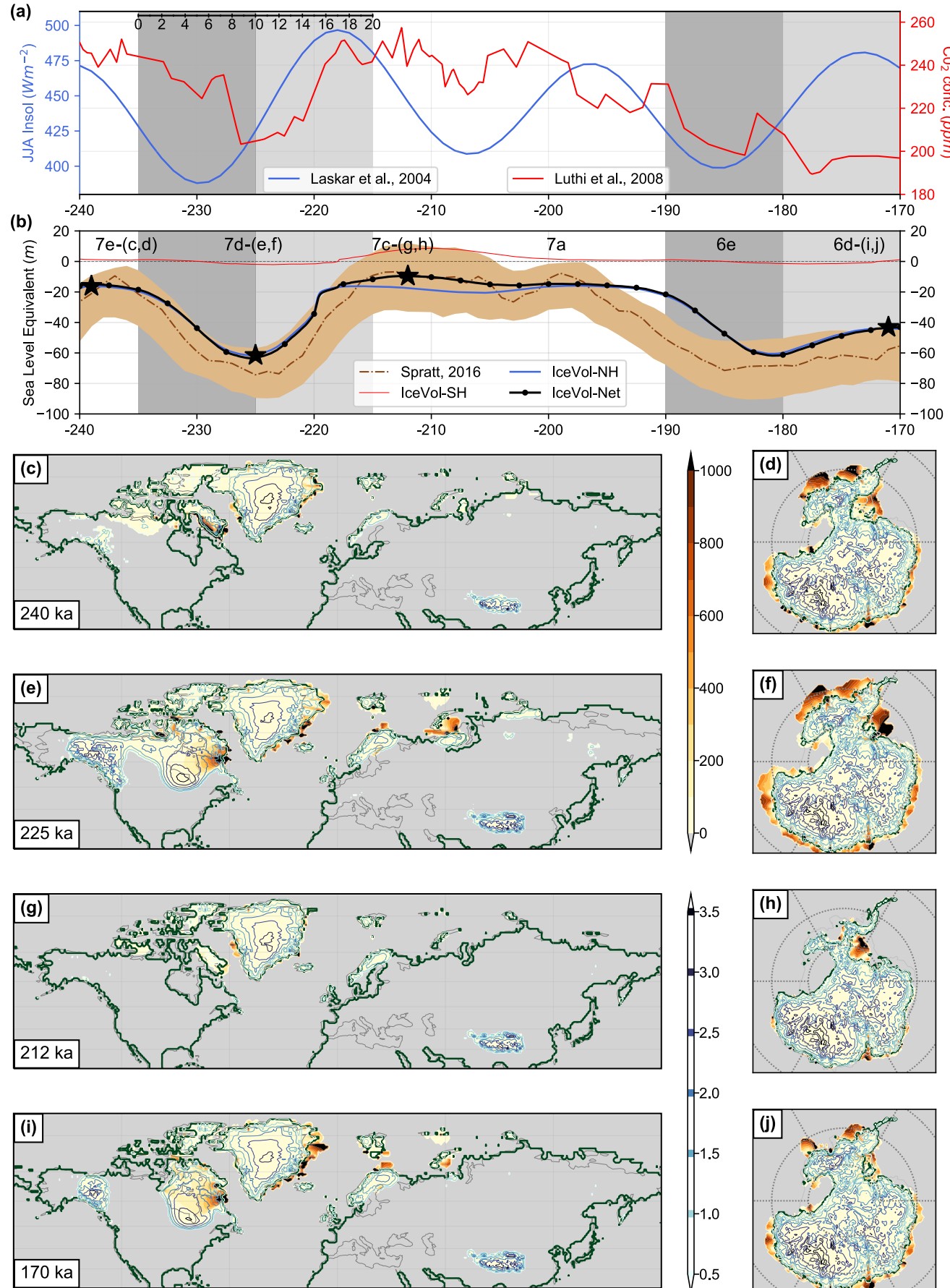

**Figure 4: Maps of ice height and ice velocity from our transient coupled climate-ice sheet simulation over MIS 7. (a) JJA mean insolation at 65°N (Wm⁻², blue, *(Laskar et al., 2004)*) and CO₂ concentration (ppm, red *(Lüthi et al., 2008)*). The dark grey and light grey patches in the backgrounds of (a) and (b) refer to two 20ky periods leading into a glacial inception (10ky) and immediately after a glacial inception (10ky). The duration of these periods (20ky) is marked by a small scale on the top left and used in Figure 6. (b) Global sea level estimates (m) from *Spratt and Lisiecki (2016)* (brown) and sea level equivalent of ice volume from SH (red), NH**

(blue) and total (black) from our transient simulation. The marked stars on the simulated SLE represent four instances corresponding to MIS 7e (240ka), MIS 7d (225ka), MIS 7c (212ka) and MIS 6 (170ka). (c) Basal ice velocity (solid colors, my$^{-1}$); ice thickness (colored contours, km) and the grounding line (solid green lines) for the Northern Hemisphere at 240ka, initial condition. (d) Same as (c) but for the southern hemisphere. (e) & (f) Same as (c) and (d), but for 225 ka. (g) & (h) Same as (c) and (d), but for 212 ka. (i) & (j) Same as (c) and (d), but for 170 ka.

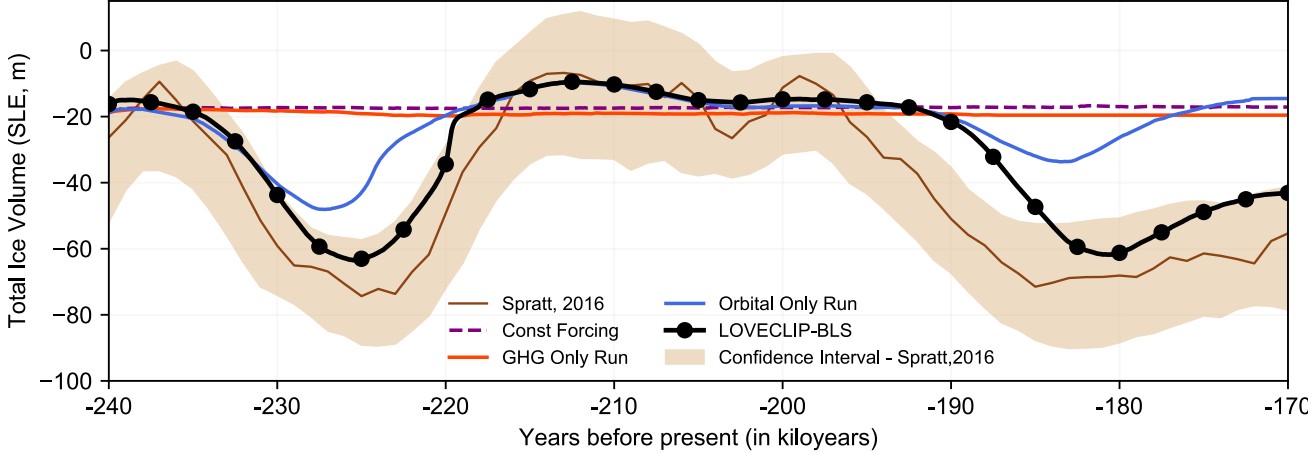

**Figure 5: Effects of orbital and GHG forcings on simulated ice volumes during MIS 7.** Sea level reconstruction (m) and 95% confidence interval of Spratt and Lisiecki (2016) (brown). Total ice volume (in terms of SLE, m) from transient LOVECLIP simulations with: (i) constant orbital and GHG values set at 240ka (dashed purple line); (ii) orbital values set at 240ka but time varying GHG values (red); (iii) time varying orbital values with GHG values set at 240ka (blue); and (iv) time varying orbital and GHG values (black marked, BLS). All experiments are conducted with an α value of 2 and $m$ of 125 Wm$^{-2}$.

845

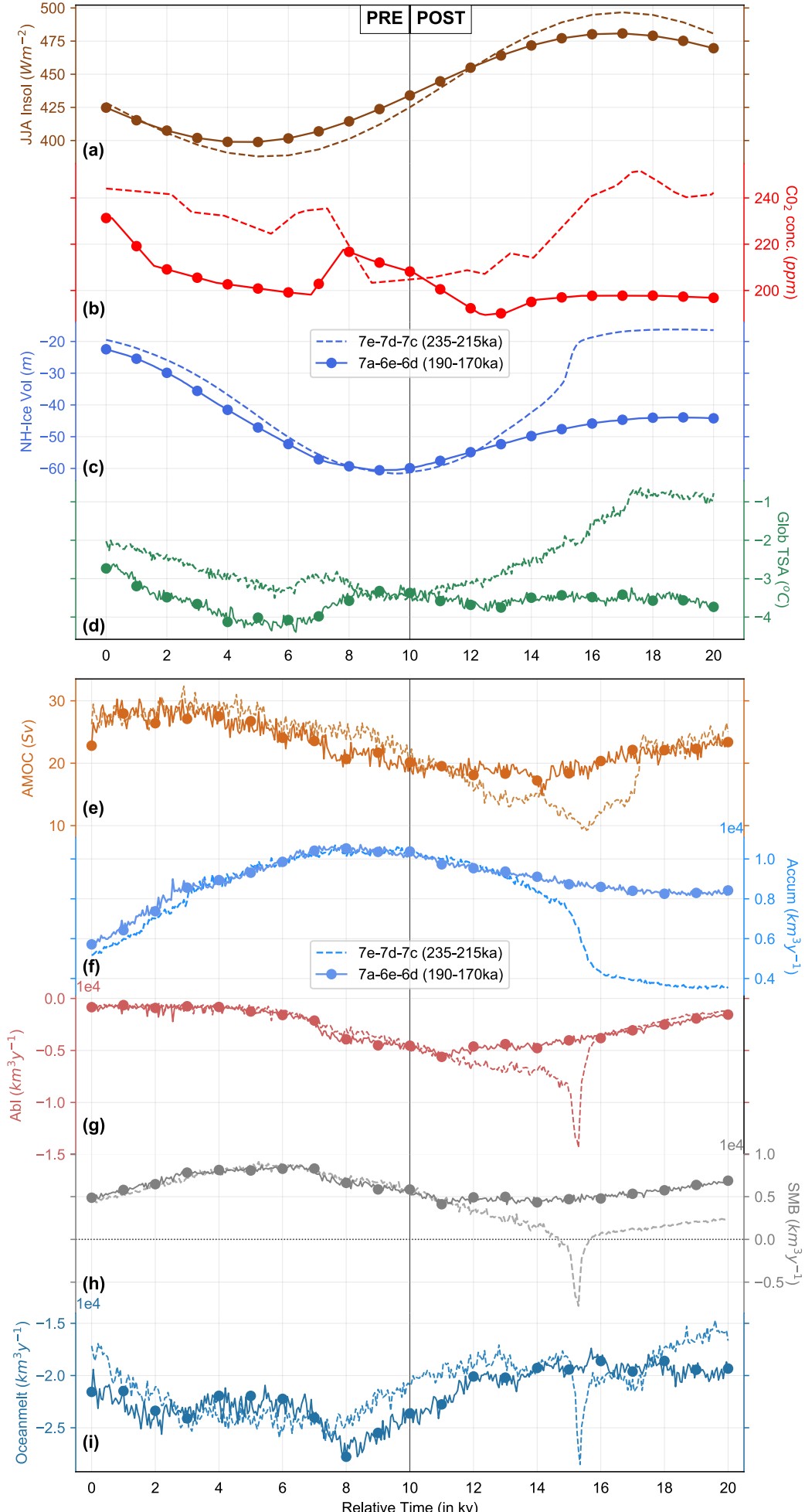

**Figure 6: Comparison between two glacial inception scenarios. Different variables over two 20k year periods (relative time) during pre-inception (left half) and post inception (right half) over the Northern Hemisphere are plotted. Variables from the earlier period (235-215ka) are plotted in dashed lines while that of the later period (190-170ka) are plotted in circled solid lines. (a) JJA mean insolation at 65˚N (Wm$^{-2}$, _(Laskar et al., 2004)_). (b) $CO_2$ concentration (ppm, _(Lüthi et al., 2008)_). (c) Simulated Northern Hemisphere ice volume, in sea level equivalents (m). (d) Global average surface temperature anomaly (˚C). (e) Temporal evolution of AMOC (Sv). (f) Net integrated accumulation rate over Northern Hemisphere ice (km$^3$y$^{-1}$). (g) Net integrated surface melt rate over Northern Hemisphere ice (km$^3$y$^{-1}$). (h) Net integrated surface mass balance over the Northern Hemisphere ice (km$^3$y$^{-1}$). (i) Net integrated subshelf melt rate over Northern Hemisphere (km$^3$y$^{-1}$).**

855

860

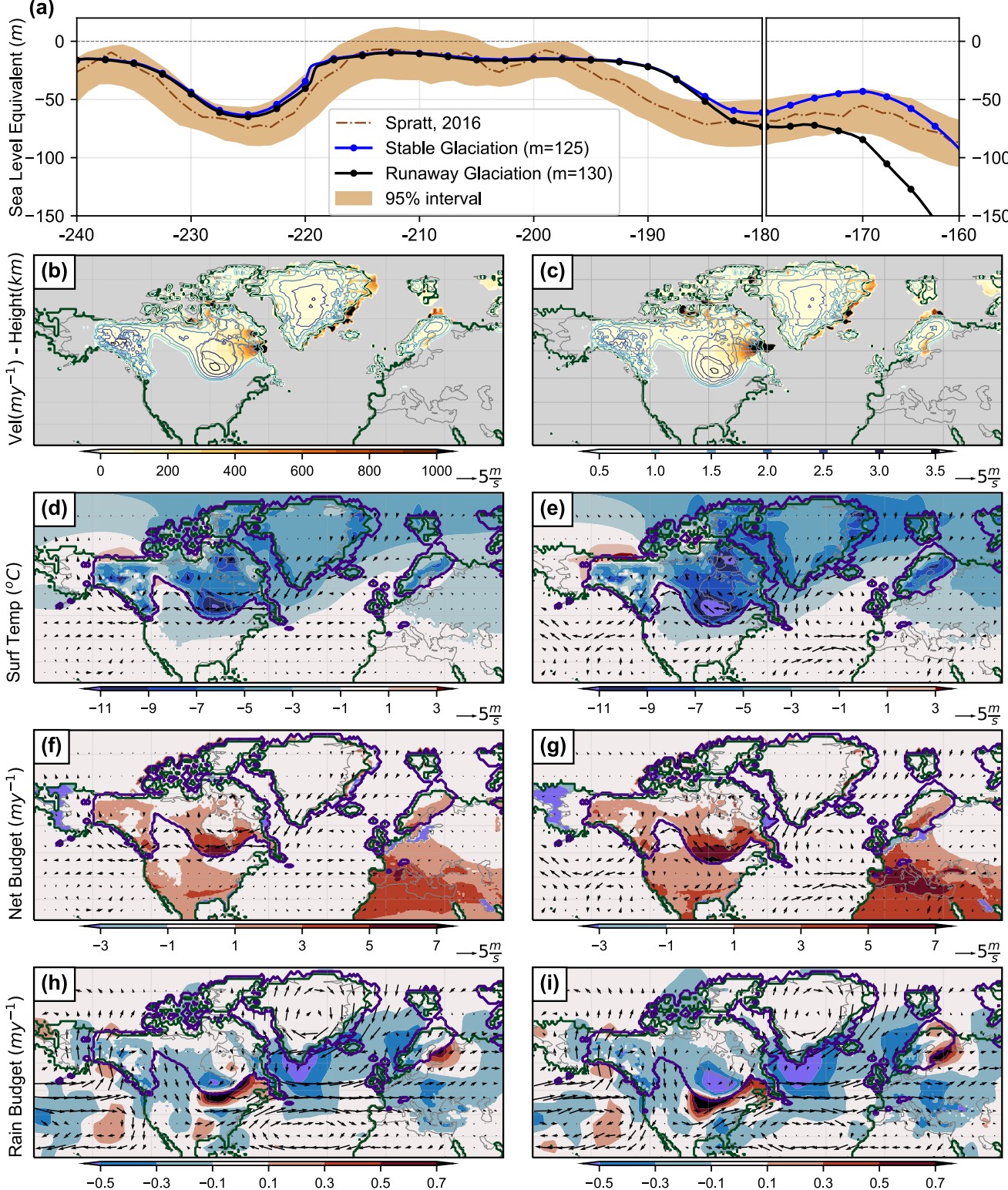

**Figure 7: Bifurcation of the system at 180ka while transitioning into MIS 6 over Laurentide. (a) Sea level reconstruction (m) and 95% confidence interval of Spratt and Lisiecki (2016) (brown). Total ice volume (in terms of SLE, m) from two ensemble members of LOVECLIP, one that leads to a stable glacial inception (blue; $\alpha$=2, $m$=125 Wm$^{-2}$) and another into a runaway glaciation (black; $\alpha$=2, $m$=130 Wm$^{-2}$). Climate and ice sheet variables at 180ka from the stable glaciation on the left column (b, d, f and h) and runaway glaciation on the right (c, e, g and i). (b,c) Basal ice velocity (solid colors, my$^{-1}$) overlaid with ice thickness (colored contours, km) and the grounding line (solid green lines). (d,e) Surface temperature anomalies ($^{\circ}C$) overlaid with *anomalous* wind vectors at 800hPa (ms$^{-1}$). (f,g) Net mass balance anomalies (my$^{-1}$) overlaid with *anomalous* winds (ms$^{-1}$). (h,i) Rainfall anomalies (my$^{-1}$) overlaid with *absolute* winds (ms$^{-1}$). The purple contours in (d) to (i) mark the boundaries of the ice sheets from each run (stable for left and runaway for right). Anomalies here are with respect to the initial condition at 240ka. Anomalies over the Eurasian and Siberian ice sheets are small and not shown.**