# Peer review of "Simulating Marine Isotope Stage 7 with a coupled climate-ice sheet model"

_Climate of the Past, 2020_

## Referee Comment (RC1) · Anonymous Referee #1 · 29 Apr 2020

**General comments**

Choudhury et al. present simulations of the glacial inception MIS7 (240-170 ka) using a climate model coupled to an ice sheet model (North and South). They discuss the simulated global sea level evolution during MIS7 and in particular the two step glaciation. While coupled ice sheet – climate simulations spanning several thousand years are still scarce, the paper shows very interesting results. I have listed below a few comments / suggestions that could be considered before resubmission.

**Specific comments**

1- Methods

- I found the coupling strategy not very clear. If I understand correctly, the surface mass

balance model is a submodel of PSUISM. From Fig. 2 it seems that PSUISM only takes SAT, solar radiation, precipitation and Tocean. How these fields are downscaled onto the higher resolution ice sheet model grid? Then, what is the "solar radiation" (not explained in Sec. 2.3)? I find it a bit strange that surface albedo is not part of the fields that are given to PSUISM. Eq.1 and Eq.2 suggest that PSUISM compute its own albedo from rs and rl. Where do these rs and rl come from? Also, you do not explain how you compute the sub-shelf melt from Tocean. Is this a simple scaling after a bilinear spatial interpolation? What about ice shelves developing where there is no oceanic grid points (e.g. Kara and Barents seas)? I am sure that we can find all this information in the different papers that use a similar LOVECLIP framework but it would facilitate the reading to have a synthetic and integrated view in this paper as well.

- On the coupling strategy again, but from the ice sheet to the climate model this time. I understand the use of the acceleration factor. However, I do not understand how you can ensure water conservation between the ice sheet and the rest of the climate system with this acceleration factor. It seems to me that when we use an acceleration factor we can only conserve the volume or the flux, but not the two quantities at the same time (already reported by Heinemann et al., 2014). For example, let's assume that the ice sheet model runs 10 years for each year computed by the rest of the climate model. Let's say that the ice sheet model produces 10 km3 of volume loss integrated over the 10 years (1km3/yr). What do you give to the ocean model? 1 km3 or 10 km3 ? The second option conserves the volume but the fluxes that arrive to the ocean are overestimated which can ultimately result in unrealistic oceanic evolution. In any case, I think it needs a bit more of description in the paper. Also, what is the routing strategy to transfer the ice loss to the ocean? Do you use the atmospheric model runoff model? In case of an ice sheet inception, do you have a negative flux at the surface of the ocean that increases the salinity? Is this spatially resolved?

- Have you run a pre-industrial control run (for example a 10-kyr long simulation with the LGM bathymetry under pre-industrial orbital and GHG forcing)? Such simulation

could be nice to validate your model setup, or at least to quantify the bias in the coupled model trajectory.

- By starting at 240 ka, you start towards the end of a deglaciation. Do you think that the long-term climate trajectory has an impact on your results? For example do you think that a restart from a full glacial state at 250 ka would result in a similar global ice volume evolution across MIS7?

- You use a simple bias correction for surface temperature and precipitation. Why not use a similar technique for the radiative inputs of the ice sheet model and for the oceanic temperature as well?

- The hydrofracturing and cliff collapse parametrisation embedded in PSUISM is controversial (e.g. Edwards et al., 2019). Do you think that you would end up with different ice volume trajectories using a model that does not account for the MICI?

2- Results

- From my understanding of your model results, the respective role of orbital configuration with respect to CO2 is pretty much linked to the choice of the alpha / m combination. From Fig. 3 it seems that you cannot guarantee the uniqueness of your calibrated alpha / m (higher m but lower alpha might works equally well than lower m and higher alpha). As a result how robust is your conclusion on the respective role of orbital versus CO2?

- You justify the use of the alpha parameter to correct for the lack of sensitivity of the climate model. There are alternative way to modify the climate sensitivity in the model. For example Loutre et al. (2011) modify a set of model parameters in order to have a similar pre-industrial climate but different sensitivity to the change in CO2. With your approach you might give too much importance to the radiative effect of CO2.

- The Eurasian ice sheet in your simulation does not grow at all which seems in contradiction with palaeo data (e.g. Batchelor et al., 2019). Even in the run-away glaciation

presented in Fig. S6, it seems that there is only very little ice developing in Eurasia. I know that it is not trivial to simulate satisfactorily the Eurasian ice sheet inception, particularly the Kara-Barents sector. Do you have any idea on how to improve on this? Do you think that it is an oceanic problem, e.g. too warm waters in the Kara and Barents seas? Or an atmospheric problem, e.g. shift in storm tracks?

- For the two step inception (Sec. 3.4), you claim that $CO_2$ explains the later full inception. If the insolation signal is indeed similar for both periods, it shows nonetheless a greater amplitude in MIS 7e-7d-7c with respect to MIS 7a-6e-6d. The insolation maxima is thus slightly smaller for the later period. This slight difference in the insolation maxima could explain the ice retreat in the older inception, with the full glaciation being the result of an insolation threshold? Of course $CO_2$ should help but it is hard to distinguish the role of the two forcings (especially also because the results depend on the value of the chosen alpha parameter).

- I wonder how robust are the atmospheric circulation changes discussed in Sec. 3.6. For example, don't you think that the atmospheric circulation might be potentially largely affected by the presence of an ice sheet in Eurasia?

3- Miscellaneous

- l64-78 You could refer to your figure 1 somewhere around here to facilitate the reading.

- towards l194 and Fig. S1 From my understanding of your methods your reference climate model uses a last glacial maximum bathymetry (since bathymetry is fixed to a LGM value). Would it have been appropriate to compute the bias for the modern climate using a LGM bathymetry? More generally it would be very interesting to see the effect of the sole bathymetry on the simulated climate (under PI forcing for example).

- l252 The Filchner-Ronne ice shelf is advancing but there is almost no change for the Ross ice shelf. Any idea why?

- l306-312 You show in the figure the mean SMB/ablation/accumulation over the ice

sheet. In doing the spatial average, we can end up with very similar values for very different ice sheet extent. In addition, we cannot quantify the importance of the different processes to explain the total mass change (for example the sub-shelf melt is much more negative than SMB but it may concern only a very small fraction of the ice sheet). Instead, it could be interesting to have the integrated value over the ice sheet but not divided by the extent (Gt or km3 per year), to have a better idea of the respective role of SMB and sub-shelf melting to explain the total ice volume change.

- l319-322 The spike in SMB is very impressive but I am even more surprised by the spike in shelf melting. I think it could be useful to show maps of surface mass balance and sub-shelf melting for temporal snapshots in the vicinity of this event (circa 210 ka?). In doing so, it will illustrate the saddle-collapse as well as the spike in sub-shelf melt. Again, maybe the spike in sub-shelf melt is affecting a tiny area and is not representative of the total mass loss (previous comment)?

- l345-346 Do you know the reason for this? It would be nice to understand this circulation change (which produce the spike in sub-shelf melt?). It happens during the deglaciation and thus, in the meantime, you probably have a lot of freshwater flux that is discharged to the ocean. Does the two processes (increase in sub-shelf melt and freshwater flux) are related in some way? If yes, how realistic are your freshwater fluxes (see previous comment on the acceleration factor).

- Fig. 2 You could also mention on the figure how the interpolation / downscaling to the different grids is done. Also, by "landmask" you mean "ice mask" (albedo)?

**Technical comments**

- Fig. 4 The grounding line is hardly distinguishable.

- Fig. 4 (and Fig. S2, S3, S4 and S5) I find the maps of mixed information thickness / ice velocity hard to read. Maybe using the contours for thickness and the solid colours for velocity would look nicer?

- Fig. 7 I am sure that you can use a better colour bar for panel f to i. At least it can be white where there are small changes instead of yellow.

- Fig. 7 l675 purple contours are for which run (blue or black in panel a)?

- Fig. S1 Panel b: since a multiplicative correction of 1 means no correction maybe it could have been better to have a neutral colour such as white around the value of 1.

**References**

Edwards, T. L., M. A. Brandon, G. Durand, N. R. Edwards, N. R. Golledge, P. B. Holden, I. J. Nias, A. J. Payne, C. Ritz, et A. Wernecke, Revisiting Antarctic ice loss due to marine ice-cliff instability, Nature, 566, 58–64, 2019.

Loutre, M. F., Mouchet, A., Fichefet, T., Goosse, H., Goelzer, H., and Huybrechts, P.: Evaluating climate model performance with various parameter sets using observations over the recent past, Clim. Past, 7, 511–526, https://doi.org/10.5194/cp-7-511-2011, 2011.

---

## Referee Comment (RC2) · Lev Tarasov (Referee) · 22 May 2020

This submission presents the first asynchronously coupled ice sheet and LoveClim transient modelling results for the MIS 7 to 6d (240 to 170 ka) interval. It is therefore plenty novel and relevant for CP. Two of its main conclusions reflect those of another submission currently under CPD review examining last glacial inception (Bahadory et al, cp-2020-1, of which I'm a co-author): that 1) LoveClim appropriately coupled with an ice sheet model can within uncertainties capture major interglacial-glacial-interstadial transitions and 2) replicating such transitions can impose strong constraints on the coupled model. The third conclusion, that orbital forcing has more impact than GHG changes during glacial inception, is not surprising based on associated radiative forcings and chosen fixed forcing states but does not necessarily hold when one considers say the whole last glacial cycle (Tarasov and Peltier, JGR 1997, albeit with a much simpler 2D energy balance climate model and isothermal ice sheet model).

As to the quality and limitations of the scientific methods, aka model configuration and experimental design, the component PSU ice sheet model and LoveClim EMIC are well documented and well used models. They arguably remain near state-of-the-art for transient glacial contexts (though LoveClim is sorely in need of a replacement and ongoing work with transient GCM/ISM models are defining a new state of the art for very small ensembles). However, I do find some limitations that need to be made explicit in the manuscript. As shown in Bahadory et al, GMD 2018 (again from my group and curious why this paper is not cited), accounting for orographic forcing of precipitation in downscaling to the ice sheet grid and accounting for topographic changes in meltwater routing can each have significant impact on modelled ice sheet evolution. To be concrete, inclusion of the latter, for instance can, result in more than 15 m eustatic sealevel equivalent discrepancies within 4 kyr (IBID). The current submission is not even explicitly clear if the modelled surface freshwater routing changes during the glacial cycle (though it appears not to be the case). There are also a host of other sources of model uncertainty that are not mentioned, including: length of model spinup, topographic upscaling from ISM to LOVECLIM, the requirement of much higher ice sheet resolution or subgrid mass-balance accounting to adequately represent restricted glaciation over Alaska, and the dozens of poorly constrained parameters in both models that the modellers have chosen to not vary.

The other hole in the paper for me is pervasive in paleo ice sheet modelling: a very limited exploration of the impact of model uncertainties, given the small number of ensemble parameters and limited ensemble size. This is an exploratory work, and so arguably gets a pass with this limited ensemble, but I encourage the authors to expand their set of ensemble parameters and ensemble size in future work. And the paper needs a bit more attention to discussion of uncertainties that arise from the very limited ensemble size and the potential impact thereof.

The paper structure is logical. The abstract is concise and appropriate. The language is fluent, though there are instances where precision is lacking (eg "reasonably well", cf detailed commments below) as are some important (to me at least..) details about model setup.

Once the issues above and below are addressed, I would see this submission appropriate for TC publication.

**Specific comments.**

For a range of model parameters, the simulations capture the
reconstructed evolution of global ice volume reasonably well
**What does "reasonably well" mean. Be precise**

It is demonstrated that glacial inceptions 20 are more sensitive to
orbital variations, whereas terminations from deep glacial conditions
need both orbital and greenhouse gas forcings to work in unison

**this likely depends on your choice of fixed orbital configuration**
**cf Tarasov and Peltier, JGR 1997.**

This poses a general challenge for transient coupled climate-ice sheet
modeling.
**on the flip side, it poses a strong constraint opportunity, cf**
**Bahadory et al, cp-2020-1.pdf in TCD**

which correspond to about 1.3 mm/year global sea level equivalent
during the build-up phase.
**that number is more than a factor too small for last glacial**
**inception if one goes by the cited LR04 stack**

fig 3 captions
**again mixing up ensemble with ensemble run. An ensemble is a collection**
**of model runs.**

fig 3
**does this show all the model runs in the non-fixed forcing ensemble you**
**carried out? If so, please make this clear.**

including multi-ensemble simulations
**do you mean mult-run or did you actually carry out multiple ensembles?**
**If so, how large was each ensemble?**

The effect of CO2 variations with respect to the reference CO2
concentration (365ppm) on the longwave 120 radiation flux is scaled up
by a factor $\alpha$, to account for the low default sensitivity of ECBilt to
changes in CO2 concentrations (Friedrich and Timmermann,
2020;Timmermann and Friedrich, 2016). $\alpha$ is determined based on
transient past and future simulations.
**Please provide the pCO2 ECR for alpha=2 with your setup. This would**
**let reader better judge how consistent this resultant sensitivity is**
**compared to that of IPCC grade GCMs. Also, it would be worthwhile**
**comparing your \alpha to that found based on 1D radiative-convective**
**modelling (Ramanathan et al, 1979 JGR).**

2.2 PSUIM surface mass balance description, eq 1 and 2
**on what timestep is this carried out? If longer than 1 hour**
**(presumably), what accounting is there for diurnal variations?**

 after eq 4 : with $r = max J0, min[1, (T^* + 3)/3]$
**Based on my on examination of ice sheet model horizontal basal**
**temperature between along flow adjacent grid cells (which provides**
**logical upper bound for the transition range), 3 C is a wide**

**transition range for warm based sliding. How is this justified?**

For the NH, a binary sliding coefficient map ... low sliding over
present-day land $(C(x, y))$ = ... representing non-deformable rock).
**Much of Southern Canada and Northern USA (regions of glacial ice**
**cover) is covered by tills, not hard beds and this can significantly**
**influence ice sheet evolution (eg Tarasov and Peltier, 2004**
**QSR). How do you justify making all this hard bedded?**

Preliminary experiments (not shown) with different acceleration
factors suggest that model results do not change significantly when
N <= 5.
**Please be more precise by what "significantly" means.**

Furthermore, for surface temperature $T$ , a lapse-rate correction of
8˚C km−1 is applied to account for differences between LOVECLIM
orography and PSUIM topography and precipitation is multiplied by a
Clausius–Clapeyron factor of 2^.. with $\Delta T$ being the
temperature lapse-rate correction, to account for the elevation
desertification effect (DeConto and Pollard, 2016).
**How do you justify using a lapse rate that is inconsistent with**
**the lapse rate LOVECLIM uses internally? For future work, I would**
**strongly advise inclusion of orographic forcing given the impact**
**thereof missed in a coarse grid EMIC (cf eg Bahadory**
**and Tarasov, GMD 2018)**

Basal melting and liquid runoff from PSUIM is discharged via
LOVECLIM's runoff masks in both hemispheres;
**do these masks account for changing topography? And if so, what**
**accounting is there for critical subgrid gateways for southern**
**drainage from the NA North American) ice complex (cf eg Tarasov and**
**Peltier, QSR 2006).**

Increasing the value of $m$ (Eq. (1))
**as a reader, it is a pain to flip back 5 pages to find out what m**
**is, please add a few descriptive words (surface energy offset term**
**or some such) ditto for \alpha**

3.2 Ice sheet evolution
**this section would be strengthened with more contact with the**
**(albeit limited) glacial geological litterature. The key relevant**
**data are Late Pleistocene glacial limits.  Does your model respect**
**them everywhere? If not, what are the main discrepancies?  The only**
**regions I see that could be at issue are your Alaskan incursion and**
**Northern Siberia.**

the glaciation 235 into MIS 6 is delayed by ~3ky (191ka instead of
194ka).
**Do you really believe that temporaly uncertainty in inferred**
**sealevel is < 3 kyr that far back?**

After a relatively stable interglacial state till MIS 7a, the system
moves into the next glacial and reaches a glacial equilibrium state.
**This description does not accurately reflect your figure 3, I see no**
**sign of a "glacial equilibrium"**

...Batchelor et al. (2019), have suggested a larger Eurasian ice sheet
over the MIS 6 period (160-140ka),
**"suggested" does not accurately nor precisely reflect the**
**inferences. Be more accurate: eg glacial geological record indicates**
**that the asynchronous maximal MIS 6 ice margins are outside of MIS 2**
**ice margins.**

**leading to temperatures low enough (Fig. 6d) to avoid ablation even if**
**the Laurentide extends equatorward**
There is always seasonal ablation on an northern ice sheet. Be more
precise.

Figure 7:
**makes it a lot easier for the reader if subplots have descriptive**
**headings on the plot. Having to visually jump between each subplot and a large**
**caption disrupts reader assimilation of the plots.**

Fig 7 caption two ensembles of
**do you mean two ensemble members?**

Fig 7f-i
**I find the colour scheme has insufficient and distorting colour range. Eg**
**for 7h the 0.3:0.5 colour is just a shade darker than the -0.3:-0.1 range**
**colour. Furthermore, it makes no sense that the plot has regions where**
**these colour border each other without any intermediate ranges showing.**

Fig 7:
**I am a bit confused why there is such limited glaciation east of the**
**Canadian Cordillera, given the northwesterly (and therefore relatively colder)**
**absolute winds and rainfall anomalies that match (within the colour**
**scheme) other sectors with significant ice cover.  Is this due to**
**the temperature bias correction or limited rainful or ? On that note,**
**a short discussion on the impact of the bias correction would aid**
**interpretation of its role in your results.**

this behavior is reminiscent of a saddle node bifurcation

We find that small changes in the Laurentide's ice distribution for
similar total ice volumes can lead to a saddle node 400 bifurcation of
the system
**which is correct? Have you shown this to be a saddle node bifurcation**
**or is this reminiscent of a saddle node bifurcation?**

Also, the stationary wave feedback reported here 410 could be a model
dependent feature of LOVECLIM, given it has only three atmospheric
levels
**and LOVECLIM is run at a relatively coarse T21, while the**
**litterature indicates that at least T42 is needed to avoid major**
**resolution sensitivity of the eddy driven jet (eg Lofverstrom and**
**Liakka, 2018).**

Results also suggest that our coupled simulations are realistic over a
narrow range of parameters
**what does "realistic" mean? Again, be precise**

is more difficult than conducting timeslice experiments
**I would say much more difficult and therefore offers much more**
**self-constraint**

Fig S1
**summer (JJA for NH and DJF for SH) temperature is much more critical**
**for ice sheet growth than mean annual temperature, given surface mass-balance**
**dependencies, so please add these plots.**

---

## Author Comment (AC1) · 19 Jun 2020

We thank the reviewer for their constructive comments, which we believe have helped improve the quality of the manuscript. To address their concerns, we have made substantial revisions. Specifically, we have expanded and clarified the description of the coupling mechanism between the climate and ice sheet model. We have also added more discussions of our results in the context of previous studies of reconstructions. Detailed responses to the issues raised by them can be found below along with excerpts from the revised manuscript in boxes. A bibliography of the references cited here is present at the end of the document. We would like to emphasize that excerpts presented here are only a handful of the changes made to the manuscript, which are relevant to the reviewer's distinct comments. Besides these, there are many other changes designed to improve the overall appeal of the study.

**Reviewer 1:**
**Specific comments:**

*1- Methods:*
*I found the coupling strategy not very clear. If I understand correctly, the surface mass balance model is a submodel of PSUISM. From Fig. 2 it seems that PSUISM only takes SAT, solar radiation, precipitation and Tocean. How these fields are downscaled onto the higher resolution ice sheet model grid? Then, what is the "solar radiation" (not explained in Sec. 2.3)? I find it a bit strange that surface albedo is not part of the fields that are given to PSUISM. Eq.1 and Eq.2 suggest that PSUISM compute its own albedo from rs and rl. Where do these rs and rl come from? Also, you do not explain how you compute the sub-shelf melt from Tocean. Is this a simple scaling after a bilinear spatial interpolation? What about ice shelves developing where there is no oceanic grid points (e.g. Kara and Barents seas)? I am sure that we can find all this information in the different papers that use a similar LOVECLIP framework but it would facilitate the reading to have a synthetic and integrated view in this paper as well.*

A. We have now expanded the section on the climate-ice sheet coupling to be more complete and easier to read. First, we present answers to each specific question below:
   - LOVECLIM outputs of temperature, radiation and precipitation are downscaled using bilinear remapping and ocean temperature is downscaled using a conservative remapping approach.
   - Solar radiation refers to the incoming solar radiation at the surface (used in Eq. 3 for calculating the surface mass balance).
   - PSUIM calculates albedo based on the fraction of area that is covered by snow ($r_s$) and that without ($1 - r_s$) \. For the snow-covered areas, the fraction of area corresponding to wet and dry snow is also explicitly considered using the ratio between liquid water contained in the snow mass and the maximum embedded liquid capacity ($r_l$). All of this information is obtained from the last year of its previous chunk of PSUIM run. This provides a more realistic estimate of the albedo than downscaling from LOVECLIM.
   - The sub-ice-shelf ocean melting is calculated as per Pollard et al. (2015) using ocean temperature at 400m depth ($T_o$) from LOVECLIM as follows:

   $$OM = \frac{KK_T \rho_w c_w}{\rho_i L_f} |T_{oc} - T_f| (T_{oc} - T_f) \quad\quad\quad (5)$$

   where $OM$ is the subshelf ocean melting rate ($myr^{-1}$), $T_{oc}$ is the LOVECLIM ocean temperature at 400m depth (°C), $T_f$ is the ocean freezing temperature at depth z (m) calculated as $T_f = 0.0939 - 0.057 \times 34.5 - 0.000764 \times z$, $K_T = 15.77 myr^{-1}K^{-1}$ is a coefficient; $K$ is a non-dimensional factor of order 1, $K = 3$; $\rho_w = 1028 kgm^{-3}$ is the density of ocean water; $\rho_i = 910 kgm^{-3}$ is the ice density; $c_w = 4218 Jkg^{-1}K^{-1}$ is the specific heat of ocean water; and $L_f = 0.335 \times 10^6 Jkg^{-1}$ is the latent heat of fusion .
   - $T_{oc}$, used in calculating the ocean sub-ice-shelf melting (Eq. 5) is interpolated to PSUIM grid using conservative remapping. For some of the floating ice (shelves) in PSUIM (Eq. 2), the ocean points underneath may not get an ocean temperature assigned on interpolation from LOVECLIM, since the land-sea mask in LOVECLIM is kept constant. For each of such grid points, the algorithm averages over the neighboring eight PSUIM grid points and this process is repeated until all PSUIM ocean points get ocean temperatures.
   Eq. 2 referred here is:

   $$\left. \begin{array}{l} \rho_w(S - h_b) < \rho_i h; \text{ and } h_s = h + h_b; \text{ for grounded ice (ice sheet)} \\ \rho_w(S - h_b) > \rho_i h; \text{ and } h_s = S + h\left(1 - \frac{\rho_i}{\rho_w}\right); \text{ for floating ice (ice shelves)} \end{array} \right\} \quad (2)$$

   where $\rho_w = 1028 kgm^{-3}$ is the density of ocean water; $\rho_i = 910 kgm^{-3}$ is the ice density; $S$ is the sea level ($m$); $h_b$ is the bedrock elevation ($m$); $h$ is the ice thickness ($m$); and $h_s$ is the ice surface elevation ($m$).

We have now addressed all of these issues in the model description sections of PSUIM (Sect. 2.2) and LOVECLIP (Sect. 2.3). Some relevant sections from the revised manuscript are attached below.

The criterion for grounding and floating ice is defined in *Lines 151-154* in *Section 2.2*:

> The ice is determined to be grounded or floating based on buoyancy, as per:
>
> $$\left.\begin{array}{l} \rho_w(S - h_b) < \rho_i h; \text{ and } h_s = h + h_b; \text{ for grounded ice (ice sheet)} \\ \rho_w(S - h_b) > \rho_i h; \text{ and } h_s = S + h\left(1 - \frac{\rho_i}{\rho_w}\right); \text{ for floating ice (ice shelves)} \end{array}\right\} \qquad (1)$$
>
> where $\rho_w = 1028 kgm^{-3}$ is the density of ocean water; $\rho_i = 910 kgm^{-3}$ is the ice density; $S$ is the sea level $(m)$; $h_b$ is the bedrock elevation $(m)$; $h$ is the ice thickness $(m)$; and $h_s$ is the ice surface elevation $(m)$.

Calculation of the ocean sub-ice-shelf melting is defined in *Lines 175-181* in *Section 2.2*:

> The sub-ice-shelf ocean melting is calculated as per Pollard et al. (2015) using ocean temperature at 400m depth $(T_o)$ from LOVECLIM as follows:
>
> $$OM = \frac{KK_T \rho_w c_w}{\rho_i L_f}\left|T_{oc} - T_f\right|(T_{oc} - T_f) \qquad (5)$$
>
> where $OM$ is the subshelf ocean melting rate $(myr^{-1})$, $T_{oc}$ is the LOVECLIM ocean temperature at 400m depth (°C), $T_f$ is the ocean freezing temperature at depth $z$ (m) calculated as $T_f = 0.0939 - 0.057 \times 34.5 - 0.000764 \times z$, $K_T = 15.77 myr^{-1}K^{-1}$ is a coefficient; $K$ is a non-dimensional factor of order 1, $K = 3$; $c_w = 4218 Jkg^{-1}K^{-1}$ is the specific heat of ocean water; and $L_f = 0.335 \times 10^6 Jkg^{-1}$ is the latent heat of fusion.

Downscaling of solar radiation, albedo and subshelf melting over locations without oceanic grid points in LOVECLIM are explained further in *Lines 250-257* in *Section 2.3*:

> Surface incoming shortwave radiation from LOVECLIM ($Q$, Eq. (3)) is bilinearly interpolated to PSUIM grid and then used to calculate the surface mass balance. PSUIM calculates albedo using snow covered fraction ($r_s$) and the ratio between liquid water contained in the snow mass and the maximum embedded liquid capacity ($r_l$) from the last year of its previous chunk. This provides a more realistic estimate of the albedo than downscaling from LOVECLIM. $T_{oc}$, used in calculating the ocean sub-ice-shelf melting (Eq. 5) is interpolated to PSUIM grid using conservative remapping. For some of the floating ice (shelves) in PSUIM (Eq. 2), the ocean points underneath may not get an ocean temperature assigned on interpolation from LOVECLIM, since the land-sea mask in LOVECLIM is kept constant. For each of such grid points, the algorithm averages over the neighboring eight PSUIM grid points and this process is repeated until all PSUIM ocean points get ocean temperatures.

*On the coupling strategy again, but from the ice sheet to the climate model this time. I understand the use of the acceleration factor. However, I do not understand how you can ensure water conservation between the ice sheet and the rest of the climate system with this acceleration factor. It seems to me that when we use an acceleration factor we can only conserve the volume or the flux, but not the two quantities at the same time (already reported by Heinemann et al., 2014). For example, let's assume that the ice sheet model runs 10 years for each year computed by the rest of the climate model. Let's say that the ice sheet model produces 10 km3 of volume loss integrated over the 10 years (1km3/yr). What do you give to the ocean model? 1 km3 or 10 km3 ? The second option conserves the volume but the fluxes that arrive to the ocean are overestimated which can ultimately result in unrealistic oceanic evolution. In any case, I think it needs a bit more of description in the paper. Also, what is the routing strategy to transfer the ice loss to the ocean? Do you use the atmospheric model runoff model? In case of an ice sheet inception, do you have a negative flux at the surface of the ocean that increases the salinity? Is this spatially resolved?*

A.  In the current setup, we are conserving freshwater flux and not the freshwater volume. As the reviewer rightly points out, conserving the net volume instead would lead to huge fluxes into the ocean. However, as we are conserving flux, there is an underestimation of the net freshwater volume and thus could lead to unrealistic circulation changes. This meltwater flux from the ice model is dynamically routed based on PSUIM topography till it reaches the ocean or the domain edge, and then is routed to the nearest ocean grid point in LOVECLIM. During an inception, since the precipitation is used for building ice, the runoff from PSUIM into LOVECLIM is reduced, which in turn increases the salinity in the ocean. This salinity change is spatially resolved.

We have now added discussions regarding this in *Lines 263-273* in *Section 2.3*:

The total meltwater from basal melting and liquid runoff in PSUIM is dynamically routed based on PSUIM topography till it reaches the ocean or the domain edge, and then is routed to the nearest ocean grid point in LOVECLIM. The calving flux is channeled into CLIO's iceberg model (Schloesser et al., 2019;Jongma et al., 2009) in the Southern Hemisphere (SH) and as an iceberg melt flux (freshwater flux and heat flux) in the NH (Schloesser et al., 2019). While both freshwater flux and freshwater volume cannot be simultaneously conserved in an accelerated run (Heinemann et al., 2014), we conserve freshwater flux in the current setup. The primary rational for this being that surface freshwater and meltwater fluxes are balanced by the convergence of ocean salt fluxes in equilibrium states, and adjustments of the ocean circulation are thought to occur rapidly compared to those of ice sheets. PSUIM is forced by LOVECLIM precipitation and LOVECLIM by PSUIM runoff. During a glacial inception (termination), the runoff from PSUIM into LOVECLIM reduces (increases) as ice sheets grow. This reduction (increase) in runoff into the ocean, relative to the evaporation, increases (decreases) the salinity of the ocean. This salinity change is spatially resolved.

*Have you run a pre-industrial control run (for example a 10-kyr long simulation with the LGM bathymetry under pre-industrial orbital and GHG forcing)? Such simulation could be nice to validate your model setup, or at least to quantify the bias in the coupled model trajectory.*

A.  We ran a pre-industrial control run with constant forcings set at 1850 with LGM bathymetry. The figure is attached below for reference.

[Figure]

**Figure R1: LOVECLIP simulation forced with constant 1850 forcing and LGM bathymetry. (a) Simulated ice volumes over Greenland (black) and Antarctica (red) in sea level equivalents (m). (b) Equilibrium basal ice velocity (solid colors, my⁻¹) and ice thickness (colored contours, km) for NH. (c) Same as (b) but for SH. (d) Equilibrium subsurface ocean temperature at 400m depth simulated by LOVECLIM. (e) Subsurface ocean temperature difference between LOVECLIM and WOA (Locarnini et al., 2013) dataset.**

We have clarified in the text now that the biases are based on present day simulations using an LGM bathymetry and that we do not find large-scale differences in the biases when running with present day bathymetry. This is addressed in *Lines 228-230* in *Section 2.3*:

These LOVECLIM biases are calculated for PD simulations using an LGM bathymetry. We did compare the biases between using a PD or LGM bathymetry, and while there were regional differences, the large-scale structure was found to be similar (not shown).

*By starting at 240 ka, you start towards the end of a deglaciation. Do you think that the long-term climate trajectory has an impact on your results? For example do you think that a restart from a full glacial state at 250 ka would result in a similar global ice volume evolution across MIS7?*

A. We agree with the reviewer's comment that our simulated evolution could depend on the initial condition. To restart the model from 250ka with realistic ice volumes, we would need initial conditions at that time that comes from re-running the whole setup from the previous interglacial, MIS 9 (~320ka). If we get a realistic inception and the re-adjusted parameter set can simulate a termination into MIS 7e, then we expect to get a similar ice volume evolution across MIS 7. We fully acknowledge that our results are dependent on the parameter choices and are sensitive to the initial conditions. We have now addressed this in *Lines 543-545* in *Section 4*:

> The simulated ice sheet volume is well within the range of reconstructions for a rather narrow range of parameters. Small changes in parameter values can produce strongly diverging trajectories, and the emergence of multiple equilibrium states may also suggest the model's dependence on initial conditions.

*You use a simple bias correction for surface temperature and precipitation. Why not use a similar technique for the radiative inputs of the ice sheet model and for the oceanic temperature as well?*

A. Thank you for this suggestion. We agree that including additional bias corrections as suggested above might further improve the representation of ice sheets in our model, however, neither radiation nor ocean temperature bias correction is implemented in the model version used for this study. We have recently started experimenting with implementing ocean temperature bias correction, and found that this helps simulating a somewhat more realistic, present-day Antarctic ice sheet. The effects on Northern Hemisphere ice sheets appear to be relatively smaller, presumably because of smaller coastal ocean temperature biases (e.g., Fig. R1). We have included this in the discussion for future improvements in *Lines 561-569* in *Section 4*:

> Nevertheless, there is scope of further improving the current setup. For instance, we only implement temperature and precipitation bias corrections in the current setup, and including bias corrections for radiation and ocean temperature might improve our representation of ice sheets. Future research might further improve the current setup by including the advective precipitation downscaling scheme (Bahadory and Tarasov, 2018) to account for orographic forcing, which is not captured in LOVECLIM. We are also investigating the possibilities of using a dynamical, an altitude-dependent and a $CO_2$-dependent lapse rate corrections while downscaling temperature from LOVECLIM to PSUIM. This is because the atmospheric lapse rate depends on the atmospheric $CO_2$ concentration – an effect that has not been considered so far in glacial dynamics. Furthermore, improving our basal sliding coefficient map for the NH using information of sediment sizes, instead of simply using a binary coefficient map, has the potential of further improving the simulations.

*The hydrofracturing and cliff collapse parametrisation embedded in PSUISM is controversial (e.g. Edwards et al., 2019). Do you think that you would end up with different ice volume trajectories using a model that does not account for the MICI?*

A. The contributions from hydrofracturing and cliff instabilities were negligible in our simulations and hence were not presented in the mass balance figure. Just to be sure, we repeated our analysis with hydrofracturing and cliff melt inactive in PSUIM and found no change in our ice evolutions. This is mentioned in *Lines 320-322* in *Section 3.1*:

> Following the concerns raised in Edwards et al. (2019), we ran a BLS simulation with hydrofracturing and cliff instability inactive and found no differences in the ice evolutions. This finding illustrates the complexity of the task to better constrain associated parameters in comparison of paleo climate simulations and data.

We also added discussions on the implication of these parameterizations for future studies in *Lines 546-551* in *Section 4*:

> In this context, we note that parameterizations associated with hydrofracturing and cliff instability did not impact our ice sheet trajectories. These processes have provided substantial contributions to the rapid Antarctic ice sheet retreat simulated in response to future climate projections (DeConto and Pollard, 2016), and better constraining these parameterizations is important to reduce uncertainties related to future sea level trajectories (e.g., Edwards et al., 2019). Presumably, these processes did not play an important role in our present simulations, because the climate is generally too cold, suggesting that opportunities for constraining these parameters in glacial simulations may be limited.

**2- Results:**

*From my understanding of your model results, the respective role of orbital configuration with respect to CO2 is pretty much linked to the choice of the alpha / m combination. From Fig. 3 it seems that you cannot guarantee the uniqueness of your calibrated alpha / m (higher m but lower alpha might works equally well than lower m and higher alpha). As a result how robust is your conclusion on the respective role of orbital versus CO2?*

A. Although we agree that results presented here could be influenced by parameter values, we believe our results to be robust over this time period. While Figure 3 showed only a handful of ensembles that best explain the effects of the '$\alpha$' and '$m$' parameters, the Fig. S3 (underneath) shows the whole ensemble of simulations considered in the study. These ensembles correspond to different values of '$\alpha$', '$m$', acceleration factor, $N_A = T_P/T_L$ (PSUIM chunk length corresponding to LOVECLIM chunk length), and the maps of sliding coefficient '$C$', in Eq. (6) and Eq. (7):

$$\widetilde{u_b} = C'|\tau_b|^{\mu-1}\widetilde{\tau_b} \, , \qquad\qquad (6)$$

where $\widetilde{u_b}$ is the basal sliding velocity, $\widetilde{\tau_b}$ is the basal stress; $\mu$ is the basal sliding exponent (=2); $C'$ is the basal sliding coefficient which is a function of the basal homologous temperature:

$$C' = (1-r)C_{froz} + rC(x,y) \, , \qquad\qquad (7)$$

with $r = max[0, min[1, (T_b + 3)/3]]$; where $T_b(°C)$ is the basal homologous temperature relative to the pressure melting point ($T_m = -0.000866h$, $h$ being the ice thickness in $m$); and $C_{froz} = 10^{-20} \mathrm{m \, yr^{-1} Pa^{-2}}$ (which cannot be zero to avoid numerical inconsistencies but is small enough to allow essentially no sliding).

We tried different combinations of all the parameter sets and have updated Table 1 to show all the ensemble runs performed in this study. While we performed a total of 50 separate experiments, we reported only 15 of them in Figure 3 that best describe the parameter sensitivities. We have now clarified this in in *Lines 293-300* in *Section 2.4:*

> Furthermore, sensitivity experiments with different GHG sensitivities ($\alpha$, Sect. 2.1) and melt parameterizations ($m$, Sect. 2.2) are run with full forcing. Generally, higher $\alpha$ leads to a stronger sensitivity to $CO_2$ concentrations, and higher values of $m$ strengthen buildup and weaken melting of ice during interglacial climates. These experiments are presented in the first row of Table 1 (1-15) and Fig. 3. Additional simulations with different combinations of acceleration ($N_A$), GHG sensitivity ($\alpha$), melt parameter ($m$), basal sliding coefficient maps over the NH ($C(x,y)$) and higher ice model resolution ($0.5 \times 0.25°$ for NH, $20 \times 20$ km polar stereographic for Antarctica) have been performed (experiments 16-50 in Table 1). The whole ensemble of simulations is presented in Fig. S3. Although we note that these experiments do not present a systematic evaluation of the full parameter space, ice sheet trajectories are consistent with and thereby support the conclusions presented in this paper.

The updated *Table 1* is as below:

| Expt Number | Orb Forced | GHG Forced | $N_A$ | $\alpha$ | $m$ (Wm$^{-2}$) | $C$ (myr$^{-1}$Pa$^{-2}$)-NH |
|---|---|---|---|---|---|---|
| 1 (**BLS**) | **Y** | **Y** | **5** | **2** | **125** | |
| 2 | **N** | **N** | 5 | 2 | 125 | |
| 3 | **Y** | **N** | 5 | 2 | 125 | |
| 4 | **N** | **Y** | 5 | 2 | 125 | Binary distribution |
| 5 | **Y** | **Y** | 5 | 2 | 125 | |
| 6 | **Y** | **Y** | 5 | **1.8** | 125 | **1.**Ocean: |
| 7 | **Y** | **Y** | 5 | **2.2** | 125 | $C(x,y)$=10$^{-6}$; |
| 8 | **Y** | **Y** | 5 | **2.5** | 125 | representing |
| 9 | **Y** | **Y** | 5 | **3** | 125 | deformable sediments |
| 10 | **Y** | **Y** | 5 | 2 | **80** | **2.**Land: |
| 11 | **Y** | **Y** | 5 | 2 | **100** | $C(x,y)$=10$^{-10}$; |
| 12 | **Y** | **Y** | 5 | 2 | **120** | representing non- |
| 13 | **Y** | **Y** | 5 | 2 | **130** | deformable rock. |
| 14 | **Y** | **Y** | 5 | 2 | **140** | |
| 15 | **Y** | **Y** | 5 | 2 | **150** | |
| *16-20* | *Y* | *Y* | *5* | *1.5* | *120,125,130,140,150* | |
| *21-24* | *Y* | *Y* | *5* | *3.5* | *80,100,120,125* | |
| *25-27* | *Y* | *Y* | *1 (30ky run)* | *2* | *110,120,130* | *Binary* |
| *28-30* | *Y* | *Y* | *2 (30ky run)* | *2* | *110,120,130* | |
| *31-33* | *Y* | *Y* | *10* | *2* | *110,130,150* | |

| | | | | | | |
|---|---|---|---|---|---|---|
| *34-36* | *Y* | *Y* | **10** | **2.5** | **110,120,130** | |
| *37-38* | *Y* | *Y* | **20** | **2.5, 3** | *125* | |
| *39-41* | *Y* | *Y* | *5* | *2* | *125* | *Tertiary*
***1.****Ocean:*
$C(x,y)=10^{-6}$; |
| *42-44* | *Y* | *Y* | *5* | *2* | ***150*** | ***2.1*** *Land (soft tills):*
$C(x,y)=$**$10^{-7},10^{-8},10^{-9}$**
*over northeastern* |
| *45-47* | *Y* | *Y* | *5* | ***2.5*** | *125* | *North America*
***2.2*** *Land (hard bed):*
$C(x,y)=10^{-10}$ |
| *High Resolution Runs:* $0.5 \times 0.25°$ *for NH,* $20 \times 20$ km *polar stereographic for Antarctica* | | | | | | |
| *48-50* | *Y* | *Y* | *5* | *2* | ***110,130,150*** | *Binary* |

**Table 1: List of all ensemble runs performed for the study study (shown in Fig. S3). The first 15 experiments are discussed in Sect. 3.1 and shown in Fig. 3. Values in bold represent the difference from the baseline simulation (BLS, experiment number 1). $N_A$ represents the PSUIM vs LOVECLIM acceleration factor (Sect 2.3). α represents the GHG sensitivity scaling factor (Eq. 1, Sect. 2.1) and $m$ represents the constant parameter in the surface energy balance equation (Eq. 3, Sect. 2.2). $C$ represents the basal sliding coefficient map used for the NH (Eq. 7, Sect. 2.2). All experiments are run at $1 \times 0.5°$ resolution for the Northern Hemisphere and $40 \times 40$ km polar stereographic resolution for Antarctica. The experiments in italics (16-50) are not presented here but were also performed to better constrain the parameter sensitivities.**

The whole ensemble of experiments is shown under in Fig. S3 (in supplementary):

[Figure]

**Figure S3: Transient LOVECLIP ensemble simulations over MIS7 with varying GHG sensitivities (α = 1.5-3.5), energy balance parameter ($m$ = 80-150Wm$^{-2}$), basal sliding coefficient ($C$ = $10^{-6}$-$10^{-8}$ myr$^{-1}$Pa$^{-2}$) and PSUIM-vs-LOVECLIM acceleration factor ($N_A$ = 1,2,5,10,20). The best results are obtained for α=2, $m$=125 Wm$^{-2}$, binary sliding map (ocean: $C$=$10^{-6}$ myr$^{-1}$Pa$^{-2}$ and land: $C$=$10^{-8}$ myr$^{-1}$Pa$^{-2}$) and $N_A$=5 (experiment 1 in Table 1, BLS).**

Although we could get the model to glaciate from MIS 7e to MIS 7d for a low '*α*' and high '*m*', the model does not terminate from the glaciation of MIS 7d.

Further, to verify our result that glacial inceptions are driven primarily by orbital forcings, we performed orbital-only simulations with $CO_2$ values of 180,200,220,240,260 and 280 ppmv in Fig. R2. These runs are similar to the blue line in Figure 5 (and the green line in Fig. S3 above), but with different background $CO_2$ values instead of keeping $CO_2$ constant at 240ka values (~245 ppmv). The results suggest that we can get a glacial inception over MIS 7e-7d, even for a high $CO_2$ value of 280 ppmv. Thus, we think our conclusion of glacial inceptions being primarily driven by orbital forcings is robust over this period.

[Figure]

**Figure R2: LOVECLIP simulations forced with transient orbital forcings, but constant background CO₂ values. All experiments are conducted with an α value of 2 and *m* of 125 Wm⁻².**

*You justify the use of the alpha parameter to correct for the lack of sensitivity of the climate model. There are alternative way to modify the climate sensitivity in the model. For example Loutre et al. (2011) modify a set of model parameters in order to have a similar pre-industrial climate but different sensitivity to the change in CO2. With your approach you might give too much importance to the radiative effect of CO2.*

A.  We agree that climate sensitivity could depend on responses to many different forcings. Here, we use a simple alpha parameter to change only the longwave sensitivity to CO₂, and not to other greenhouse gases. We have acknowledged this in *Lines 122-134* in *Section 2.1*:

> The effect of CO₂ variations with respect to the reference CO₂ concentration (356 ppm) on the longwave radiation flux is scaled up by a factor $\alpha$ (Eq. 1), to account for the low default sensitivity of ECBilt to changes in CO₂ concentrations (Friedrich and Timmermann, 2020;Timmermann and Friedrich, 2016;Timm et al., 2010). The effect of CO₂ on the longwave radiation is given as:
>
> $$LWR = \alpha . a(\lambda, \phi, p, t_{season}) . \log\left[\frac{CO_2(t)}{CO_2^{ref}}\right] \tag{1}$$
>
> where $LWR$ is the longwave radiation flux from CO₂; $\alpha$ is our scaling factor for the transfer coefficient, $a$, which is a function of longitude, latitude, height and season; $CO_2^{ref}$ is the reference CO₂ value set at 356 ppm. $\alpha$ changes the sensitivity of our model. For reference, the equilibrium climate sensitivity for CO₂ doubling is 3.69K for $\alpha$ of 2. Climate sensitivity is a non-trivial measure that can be changed in many different ways. For instance, changing the cloud parameterization or surface parameters would change both the longwave and shortwave forcings. Adjusting multiple

parameters may not necessarily lead to more realistic simulations. While it is possible that the climate sensitivity to some of the other forcings are also weak (Timm and Timmermann, 2007), we use a simple alpha ($\alpha$) parameter to change only the longwave sensitivity to $CO_2$, and not to other greenhouse gases. $\alpha$ is determined based on transient past and future simulations.

*The Eurasian ice sheet in your simulation does not grow at all which seems in contra- diction with palaeo data (e.g. Batchelor et al., 2019). Even in the run-away glaciation presented in Fig. S6, it seems that there is only very little ice developing in Eurasia. I know that it is not trivial to simulate satisfactorily the Eurasian ice sheet inception, particularly the Kara-Barents sector. Do you have any idea on how to improve on this? Do you think that it is an oceanic problem, e.g. too warm waters in the Kara and Barents seas? Or an atmospheric problem, e.g. shift in storm tracks?*

A. We think this to be primarily an atmospheric problem because of the coarse T21 resolution and 3-layered atmosphere of LOVECLIM; and the best way to improve on this would be to use a model with more realistic atmosphere. We have now added more discussions on our ice sheet distribution in *Lines 344-376* in *Section 3.2*:

[revised manuscript text omitted]

While the runaway simulation in Fig. S6 (Fig. S9 in the revised version) shows a very small Eurasian ice sheet, our Eurasian ice sheet grows once Laurentide growth slows down for ice volumes > 200m SLE (Fig. R3). This suggests that the relatively small Eurasian may not result from large biases in the Barents-Kara sector (also see subsurface temperature biases in Fig. R1 above), but from the timing of Laurentide and Eurasian growth alongside changes in atmospheric patterns.

[Figure]

**Figure R3: Transient LOVECLIP simulation with $\alpha$=2 and $m$=130 Wm$^{-2}$ (runaway simulation). (a) Sea level reconstruction (m) and 95% confidence interval of Spratt and Lisiecki (2016) (brown). Total ice volume from LOVECLIP run in black (SLE, m). The star shows ice volume at 152ka corresponding to 225m. (b) Basal ice velocity (solid colors, my$^{-1}$); ice thickness (colored contours, km) and the grounding line (solid green lines) for the Northern Hemisphere at 152ka. (c) Same as (b) but for Southern Hemisphere.**

*For the two step inception (Sec. 3.4), you claim that CO2 explains the later full inception. If the insolation signal is indeed similar for both periods, it shows nonetheless a greater amplitude in MIS 7e-7d-7c with respect to MIS 7a-6e-6d. The insolation maxima is thus slightly smaller for the later period. This slight difference in the insolation maxima could explain the ice retreat in the older inception, with the full glaciation being the result of an insolation threshold? Of course CO2 should help but it is hard to distinguish the role of the two forcings (especially also because the results depend on the value of the chosen alpha parameter).*

A. We agree with the reviewer that the insolation maximum is indeed smaller for the later period (Figure 6a). But we believe $CO_2$ to drive the later inception because our orbital only run shows a much weaker inception and subsequent termination over MIS 7a-6e-6d compared to MIS 7e-7d-7c in Figure 5. We have now clarified this in *Lines 408-415* in *Section 3.4*:

In spite of orbital forcing being similar over the second half of the composite figure, model trajectories diverge markedly. The successful glacial termination coincides with increasing $CO_2$ concentrations (dashed line), whereas the aborted termination during MIS 6e-6d (180-170ka, solid line) is associated with flat-lined glacial $CO_2$ values (Fig. 6a-6c). Although the insolation maximum is weaker for the later period (MIS 7a-6e-6d) compared to the earlier period (MIS 7e-7d-7c), we argue the full inception of the later period to be primarily driven by changes in $CO_2$. This is further supported by the orbital-only run showing a substantially weaker inception and subsequent termination over MIS 7a-6e-6d compared to MIS 7e-7d-7c (Figure 5), suggesting that the latter full inception could not be attributed only to an insolation threshold.

*I wonder how robust are the atmospheric circulation changes discussed in Sec. 3.6. For example, don't you think that the atmospheric circulation might be potentially largely affected by the presence of an ice sheet in Eurasia?*

A.  A high pressure cold anticyclonic pattern over ice sheets such as the Laurentide has previously been reported and was shown to be robust by Roe and Lindzen (2001). They suggest a balance between precipitation maxima alongside increased melting over the south western edges from warmer winds upslope and reduced precipitation over the rest of the ice sheet due to cold northerly winds can lead to either an equilibrium ice sheet configuration or runaways in either directions. However, it goes without saying that such patterns and runaways could change when using more realistic atmospheric models. While the presence of a Eurasian ice sheet would definitely affect the atmospheric circulation, we believe the Laurentide developing rapidly in our simulation changes the atmospheric circulation causing some warming over Europe and changed storm tracks that prevent the growth of a Eurasian ice sheet. As mentioned in the response to a previous question, our model simulates a Eurasian ice sheet only when the Laurentide growth slows down. While the circulation changes reported here maybe model dependent, Lofverstrom and Liakka (2018) reported that at least a T42 grid was needed in their atmospheric model (CAM3) to generate a Eurasian ice sheet using SICOPOLIS, albeit for the LGM. We have now mentioned this in *Lines 488-493* in *Section 3.6*:

> Roe and Lindzen (2001) suggested that the topography of an ice sheet such as the Laurentide induces a high-pressure anticyclonic circulation over the western end of the ice sheet. The associated cooling and upslope flow lead to enhanced rainfall over the western and southwestern ends of Laurentide. However, the prevailing cold northerlies downslope cause a reduction in rainfall over the ice sheet. This interplay between cooling associated with anticyclonic circulations alongside enhanced rainfall over western Laurentide and the reduction in rainfall over most of the ice sheet due to cold northerlies, can lead to an equilibrium ice sheet configuration or ice sheet growth/decay.

and in *Lines 516-522* in *Section 3.6*:

> While these stationary wave feedbacks are similar to the ones described by Roe and Lindzen (2001) and they suggested these patterns to be robust for a range of parameters, it is possible that such circulation patterns could change when using a more realistic atmospheric model or by the presence of a Eurasian ice sheet. The atmospheric patterns strengthen over the next 10ky and the runaway run simulates ~30m SLE greater ice volume than the stable run (Fig. S9). The difference between the two runs at 180ka and 170ka are presented in Fig. S10. As mentioned earlier in Sect. 3.2, it is important to acknowledge the low horizontal and vertical resolutions of LOVECLIM's atmosphere, which could mean the circulation changes reported here to be model dependent.

3- Miscellaneous:
*l64-78 You could refer to your figure 1 somewhere around here to facilitate the reading.*

A.  Done.

*towards l194 and Fig. S1 From my understanding of your methods your reference climate model uses a last glacial maximum bathymetry (since bathymetry is fixed to a LGM value). Would it have been appropriate to compute the bias for the modern climate using a LGM bathymetry? More generally it would be very interesting to see the effect of the sole bathymetry on the simulated climate (under PI forcing for example).*

A.  Yes, LOVECLIM biases are calculated for LGM bathymetry. We agree that the impact of bathymetry on climate is interesting and important. With regard to the bias correction, we did assess the impact of using bias corrections calculated for LGM vs PI bathymetry (though not for the simulations presented in this manuscript). The figure below shows temperature and precipitation bias corrections calculated for the different bathymetries, and while there are regional differences, the large-scale structure is (surprisingly) similar. While there are other issues related to the bias correction (for example, it is not obvious that the bias should be the same under LGM and PI boundary conditions), we believe that choosing LGM vs PI bathymetry does not substantially alter the results.

**LOVECLIM surface temperature and precipitation bias**

[Figure]

**a** PD - temperature        **b** LGM - temperature

**c** PD - precipitation        **d** LGM - precipitation

**Figure R4: LOVECLIM biases for present-day simulations for temperature (a) and (b); and precipitation (c) and (d). (a) and (c) use PD bathymetry, while (b) and (d) use LGM bathymetry.**

We have now clarified this in *Lines 219-233* in *Section 2.3*:

> PSUIM uses surface air temperature ($T$), precipitation ($P$), solar radiation ($Q$), and ocean temperature at 400m depth ($T_o$) as inputs from LOVECLIM. These are downscaled using a bilinear interpolation approach. The surface temperature and precipitation outputs from LOVECLIM which are used for the PSUIM surface mass balance are bias-corrected in the coupler, following Pollard and DeConto (2012), Heinemann et al. (2014) and Tigchelaar et al. (2018).
>
> $$T(t) = T_{LC}(t) + T_{obs} - T_{LC,PD} \tag{8}$$
>
> $$P(t) = P_{LC}(t) \times P_{obs}/P_{LC,PD} \tag{9}$$
>
> where $T$ is monthly surface air temperature and $P$ is monthly precipitation forcing from LOVECLIM at timestep $t$. Subscripts '$LC$', '$obs$' and '$LC,PD$' refer to LOVECLIM chunk output, observed present day climatology, and LOVECLIM present day control run, respectively. The observed present day climatology is obtained from the European Centre for Medium-Range Weather Forecasts reanalysis dataset, ERA-40 (Uppala et al., 2005). These LOVECLIM biases are calculated for PD simulations using an LGM bathymetry. We did compare the biases between using a PD or LGM bathymetry, and while there were regional differences, the large-scale structure was found to be similar (not shown). The annual mean of the monthly mean bias correction terms $T_{obs} - T_{LC,PD}$ and $P_{obs}/P_{LC,PD}$ are presented in Fig. S1. Temperature biases in LOVECLIM for boreal summer (JJA) and austral summer (DJF) are shown in Fig. S2 for reference, since summer temperatures are more crucial for ice sheet growth and decay.

*l252 The Filchner-Ronne ice shelf is advancing but there is almost no change for the Ross ice shelf. Any idea why?*

A. While our simulation does not show much variations in Antarctica, the small changes over the Filchner-Ronne shelf unlike the Ross ice shelf might be because of small differences in subsurface ocean temperature. The present study focusses more on the mechanisms underlying the response of the Northern Hemisphere ice sheets, and hence we have not performed a

detailed analysis of the processes underlying the regional response of the Antarctic ice sheets and shelves. We do agree, however, that this is an interesting and important issue, that will be further explored in future studies.

*l306-312 You show in the figure the mean SMB/ablation/accumulation over the ice sheet. In doing the spatial average, we can end up with very similar values for very different ice sheet extent. In addition, we cannot quantify the importance of the different processes to explain the total mass change (for example the sub-shelf melt is much more negative than SMB but it may concern only a very small fraction of the ice sheet). Instead, it could be interesting to have the integrated value over the ice sheet but not divided by the extent (Gt or km3 per year), to have a better idea of the respective role of SMB and sub-shelf melting to explain the total ice volume change.*

A. We have now updated Figure 6 to show integrated values of the mass balance terms:

[Figure]

**Figure 6: Comparison between two glacial inception scenarios. Different variables over two 20k year periods (relative time) during pre-inception (left half) and post inception (right half) over the Northern Hemisphere are plotted. Variables from the earlier period (235-215ka) are plotted in dashed lines while that of the later period (190-170ka) are plotted in circled solid lines. (a) JJA mean insolation at 65°N (Wm$^{-2}$, *(Laskar et al., 2004)*). (b) CO$_2$ concentration (ppm, *(Lüthi et al., 2008)*). (c) Simulated Northern Hemisphere ice volume, in sea level equivalents (m). (d) Global average surface temperature anomaly (°C). (e) Temporal evolution of AMOC (Sv). (f) Net integrated accumulation rate over Northern Hemisphere ice (km$^3$y$^{-1}$). (g) Net integrated surface melt rate over Northern Hemisphere ice (km$^3$y$^{-1}$). (h) Net integrated surface mass balance over the Northern Hemisphere ice (km$^3$y$^{-1}$). (i) Net integrated subshelf melt rate over Northern Hemisphere (km$^3$y$^{-1}$).**

*l319-322 The spike in SMB is very impressive but I am even more surprised by the spike in shelf melting. I think it could be useful to show maps of surface mass balance and sub-shelf melting for temporal snapshots in the vicinity of this event (circa 210 ka?). In doing so, it will illustrate the saddle-collapse as well as the spike in sub- shelf melt. Again, maybe the spike in sub-shelf melt is affecting a tiny area and is not representative of the total mass loss (previous comment)?*

A. The spike in subshelf melting could result from higher subsurface warming following a weakening of AMOC as reported in Liu et al. (2009) and Clark et al. (2020). We have also included a new supplementary figure, Fig. S6, showing maps of SMB and sub-shelf melting around the spikes shown in Figure 6h and 6i.

We have added the AMOC effect on subshelf melting in *Lines 418-422* in *Section 3.4*:

> For both periods the Atlantic Meridional Overturning Circulation (AMOC) is strongest during the inception phase, and weaker during termination (Fig. 6e). The AMOC weakening is substantially more pronounced in the earlier period, following the successful termination. In both periods, the AMOC recovers almost to its full interglacial state. The reduced AMOC (Fig. 6e) could lead to increased subsurface warming (Liu et al., 2009;Clark et al., 2020) causing the higher subsurface melting in Fig. 6i.

[Figure]

**Figure S6:** Temporal snapshots around the spikes in SMB and Ocean melt of Figure 6h and 6i (~15ky relative time of the dashed lines, corresponds to ~ 220ka in real time). Simulated surface mass balance (left column) and subshelf melt (right column) values at (a,b) 220ka, (c,d) 219.9ka, (e,f) 219.8ka, (g,h) 219.7ka and (I,j) 219.6ka. The blue lines mark the boundaries of the ice sheets and the black lines show the grounding line.

We have added discussions on the figure in the text in *Lines 462-469* in *Section 3.5*:

> Temporal snapshots every 0.1ky in the vicinity of the spikes in ablation (Fig. 6g), SMB (Fig. 6h) and ocean melting (Fig. 6i) are shown in Figure S6. It shows that the spikes in ablation and SMB predominantly come from a small area in the southern end of the Laurentide, while the spike in subshelf melting results only from the western part of the Laurentide with a receding grounding line. Although our model simulates sub shelf melting along the western Hudson Bay, we did not find any geologic evidence of such subsurface melting around 219.5ka. It is also worth mentioning that our setup does not simulate forebulges or other specific mechanisms modelled by more comprehensive full-Earth models. But Tigchelaar et

al. (2018) have reported such changes in mass balance arising due to changing of ice sheets to ice shelves near the grounding line.

*l345-346 Do you know the reason for this? It would be nice to understand this circulation change (which produce the spike in sub-shelf melt?). It happens during the deglaciation and thus, in the meantime, you probably have a lot of freshwater flux that is discharged to the ocean. Does the two processes (increase in sub-shelf melt and freshwater flux) are related in some way? If yes, how realistic are your freshwater fluxes (see previous comment on the acceleration factor).*

A.   We think the spike in subsurface melt to result from relatively warm ocean waters seeping in through the gap in the grounding line on the western end of Laurentide. Following the reviewer's suggestion, Fig. S6 shows the temporal snapshots of ocean melting during the spike seen in Figure 6i. The spike in subshelf melting (Fig. 6i) as well as surface ablation (Fig. 6g) during this period would definitely lead to an increase of the freshwater flux into the ocean. This would also explain the synchronization with the AMOC slowdown seen during this period in Fig. 6e. However, since the model is run at an acceleration and we are conserving the freshwater flux, the net freshwater volume dumped into the ocean would be underestimated and thus freshwater flux changes from PSUIM into LOVECLIM may not be very realistic.

We have now discussed this in *Lines 461-473* in *Section 3.5*:

Further, relatively warm subsurface ocean water (>-1°C) seeps along the west bank of Hudson Bay leading to a more pronounced negative mass balance (Fig. S5d). This shows up as a spike in the subsurface ocean melt values in Fig. 6i. Temporal snapshots every 0.1ky in the vicinity of the spikes in ablation (Fig. 6g), SMB (Fig. 6h) and ocean melting (Fig. 6i) are shown in Figure S6. It shows that the spikes in ablation and SMB predominantly come from a small area in the southern end of the Laurentide, while the spike in subshelf melting results only from the western part of the Laurentide with a receding grounding line. Although our model simulates sub shelf melting along the western Hudson Bay, we did not find any geologic evidence of such subsurface melting around 219.5ka. It is also worth mentioning that our setup does not simulate forebulges or other specific mechanisms modelled by more comprehensive full-Earth models. But Tigchelaar et al. (2018) have reported such changes in mass balance arising due to changing of ice sheets to ice shelves near the grounding line. The spike in subshelf melting (Fig. 6i) as well as surface ablation (Fig. 6g) during this period lead to an increase of the freshwater flux into the ocean. This could explain the synchronization with the AMOC slowdown seen during this period in Fig. 6e. However, since our model is run at an acceleration ($N_A = 5$) and we conserve the freshwater flux (Sec. 2.3), the total freshwater volume dumped into the ocean is being underestimated, which may distort the LOVECLIM response.

*Fig. 2 You could also mention on the figure how the interpolation / downscaling to the different grids is done. Also, by "landmask" you mean "ice mask" (albedo)?*

A.   We have clarified that we use first order conservative remapping to interpolate the data across the different grids. These are mentioned in *Line 219-220* in *Section 2.3*:

PSUIM uses surface air temperature ($T$), precipitation ($P$), solar radiation ($Q$), and ocean temperature at 400m depth ($T_o$) as inputs from LOVECLIM. These are downscaled using a bilinear interpolation approach.

"Land mask" means "ice mask" in Figure 2. We have now changed this.

**Technical comments:**
*Fig. 4 The grounding line is hardly distinguishable.*
A.   We have now changed it to solid green lines for all the figures.

*Fig. 4 (and Fig. S2, S3, S4 and S5) I find the maps of mixed information thickness / ice velocity hard to read. Maybe using the contours for thickness and the solid colours for velocity would look nicer?*
A.   Done. We have also applied the same to Figure 7, S3, S6, S6 and S8. The updated Fig. 7 is attached here as an example.

[Figure]

**Figure 7: Bifurcation of the system at 180ka while transitioning into MIS 6 over Laurentide. (a) Sea level reconstruction (m) and 95% confidence interval of Spratt and Lisiecki (2016) (brown). Total ice volume (in terms of SLE, m) from two ensemble members of LOVECLIP, one that leads to a stable glacial inception (blue; $\alpha$=2, $m$=125 Wm⁻²) and another into a runaway glaciation (black; $\alpha$=2, $m$=130 Wm⁻²). Climate and ice sheet variables at 180ka from the stable glaciation on the left column (b, d, f and h) and runaway glaciation on the right (c, e, g and i). (b,c) Basal ice velocity (solid colors, my⁻¹) overlaid with ice thickness (colored contours, km) and the grounding line (solid green lines). (d,e) Surface temperature anomalies (°C) overlaid with _anomalous_ wind vectors at 800hPa (ms⁻¹). (f,g) Net mass balance anomalies (my⁻¹) overlaid with _anomalous_ winds (ms⁻¹). (h,i) Rainfall anomalies (my⁻¹) overlaid with _absolute_ winds (ms⁻¹). The purple contours in (d) to (i) mark the boundaries of the ice sheets from each run (stable for left and runaway for right). Anomalies here are with respect to the initial condition at 240ka. Anomalies over the Eurasian and Siberian ice sheets are small and not shown.**

_Fig. 7 I am sure that you can use a better colourbar for panel f to i. At least it can be white where there are small changes instead of yellow._

A. We now use a better color scheme for Figure 7, S8 and S9. For instance, see figure above.

*Fig. 7 l675 purple contours are for which run (blue or black in panel a)?*

A. The purple contours mark the boundaries of the ice sheets simulated in each experiment, the left one is from the blue line and the right one from the black line. We have now clarified this in the figure caption (see above).

*Fig. S1 Panel b: since a multiplicative correction of 1 means no correction maybe it could have been better to have a neutral colour such as white around the value of 1.*

A. Done.

[Figure]

[Figure]

**Figure S1: Bias correction used for LOVECLIM outputs. (a) Additive bias correction for annual mean surface temperature (K). (b) Multiplicative bias correction for annual mean precipitation. Colours are log normalised for the precipitation case.**

**Reference:**

[revised manuscript text omitted]

---

## Author Comment (AC2) · 19 Jun 2020

We thank Lev for his constructive comments, which have helped improve the manuscript. To address his concerns, we have made substantial revisions. Specifically, we have expanded discussions about parameter uncertainty and included a table and figure of the entire ensemble of experiments performed. We have also added more discussions of our results in the context of previous reconstructions studies. Please find detailed responses to Lev's specific comments below, along with excerpts from the revised manuscript given in boxes. A bibliography of the references cited here is present at the end of the document. We would like to highlight that excerpts presented here are only a handful of the changes made to the manuscript, which are relevant to his specific comments. Alongside these, we have included many other changes designed to improve the clarity and readability of the study.

**Reviewer 2:**

*This submission presents the first asynchronously coupled ice sheet and LoveClim transient modelling results for the MIS 7 to 6d (240 to 170 ka) interval. It is therefore plenty novel and relevant for CP. Two of its main conclusions reflect those of another submission currently under CPD review examining last glacial inception (Bahadory et al, cp-2020-1, of which I'm a co-author): that 1) LoveClim appropriately coupled with an ice sheet model can within uncertainties capture major interglacial-glacial-interstadial transitions and 2) replicating such transitions can impose strong constraints on the coupled model. The third conclusion, that orbital forcing has more impact than GHG changes during glacial inception, is not surprising based on associated radiative forcings and chosen fixed forcing states but does not necessarily hold when one considers say the whole last glacial cycle (Tarasov and Peltier, JGR 1997, albeit with a much simpler 2D energy balance climate model and isothermal ice sheet model).*

*As to the quality and limitations of the scientific methods, aka model configuration and experimental design, the component PSU ice sheet model and LoveClim EMIC are well documented and well used models. They arguably remain near state-of-the-art for transient glacial contexts (though LoveClim is sorely in need of a replacement and ongoing work with transient GCM/ISM models are defining a new state of the art for very small ensembles). However, I do find some limitations that need to be made explicit in the manuscript. As shown in Bahadory et al, GMD 2018 (again from my group and curious why this paper is not cited), accounting for orographic forcing of precipitation in downscaling to the ice sheet grid and accounting for topographic changes in meltwater routing can each have significant impact on modelled ice sheet evolution. To be concrete, inclusion of the latter, for instance can, result in more than 15 m eustatic sealevel equivalent discrepancies within 4 kyr (IBID). The current submission is not even explicitly clear if the modelled surface freshwater routing changes during the glacial cycle (though it appears not to be the case). There are also a host of other sources of model uncertainty that are not mentioned, including: length of model spinup, topographic upscaling from ISM to LOVECLIM, the requirement of much higher ice sheet resolution or subgrid mass-balance accounting to adequately represent restricted glaciation over Alaska, and the dozens of poorly constrained parameters in both models that the modellers have chosen to not vary.*

A.  We share Lev's concern that there remains a number of parameters which are not well constrained. This is a general problem of climate-ice sheet models which are sufficiently computationally efficient to allow for simulations spanning multiple millennia, since such models cannot resolve many key processes and hence need to rely on parameterizations. On the other hand, there has been an extensive effort to constrain the parameters in the two LOVECLIP components separately, LOVECLIM and PSUIM, and sets of standard parameters have been developed, which are the default in our coupled modeling system. While it is generally possible to optimize parameter sets for a given period in time, these optimal parameter sets tend to depend on the period chosen. Presumably, this is related to model imperfections. We agree that including the processes described in Lev's comment do have the potential to reduce some of the model imperfections, however, they also come at the cost of new parameter values which need to be constrained. So, the general caveat of imperfect models and poorly constrained parameter values remains. Despite these caveats, the fact that our model is able to reproduce many transient features of past climate reconstructions gives us confidence that these models are sufficiently realistic to start using them to address questions related to physical processes and questions underlying ice sheet trajectories. As outlined below, we have revised the text to clarify the extent to which the processes above are included or not in our present model setup.

a.  Orographic forcing is not implemented in our setup. While the authors were aware of Bahadory and Tarasov (2018), it was an oversight on our part of not citing it since it was not applied to glacial simulations. We have clarified this in *Lines 244-248* in *Section 2.3*:

Instead of using a fixed value of $\gamma$, both Roche et al. (2014) and Bahadory and Tarasov (2018) used a dynamic lapse rate, where $\gamma$ is estimated locally for the ice model grids in each LOVECLIM grid. Moreover, the lapse rate also depends on the atmospheric $CO_2$ concentration. Such dynamic lapse rate corrections are not implemented in the current setup, and neither is the advective precipitation downscaling scheme of Bahadory and Tarasov (2018).

b. The modelled freshwater in PSUIM is dynamically routed based on the actual topography. We have now clarified this in *Lines 263-266* in *Section 2.3*:

The total meltwater from basal melting and liquid runoff in PSUIM is dynamically routed based on PSUIM topography till it reaches the ocean or the domain edge, and then is routed to the nearest ocean grid point in LOVECLIM. The calving flux is channeled into CLIO's iceberg model (Schloesser et al., 2019;Jongma et al., 2009) in the Southern Hemisphere (SH) and as an iceberg melt flux (freshwater flux and heat flux) in the NH (Schloesser et al., 2019).

c. Our model was spun up for 10,000 years using orbital and GHG forcings of 240ka with the NH ice volume equilibrating to -20m SLE. This is mentioned in *Lines 282-286* in *Section 2.4*:

The LOVECLIP experiments are initialized using present day ice sheet conditions and spun up using orbital and greenhouse gas (GHG) forcings of 240 ka for a period of 10ky. The model equilibrates to an ice sheet distribution in the NH corresponding to -20m SLE, implying an open Bering Strait. Our initial ice sheet distribution at 240ka is shown in Fig. 4c and is in close agreement with that used by previous studies such as Colleoni and Liakka (2020) for 239ka and Colleoni et al. (2014) for 236ka.

d. The topography from PSUIM is upscaled to LOVECLIM using a simple weighted average. We have clarified this in *Lines 259-260* in *Section 2.3*:

LOVECLIM orography and surface ice mask are updated based on the evolution of ice sheets and bedrock elevation from PSUIM. The PSUIM topography is upscaled to LOVECLIM grid using simple weighted averaging.

e. The need for higher resolution to model mass balances and transport over regions with large sub-grid relief is a very valid concern, as mentioned in Le Morzadec et al. (2015). Coarse grid resolutions average out large sub-grid relief (tall peaks and low valleys) over some mountainous regions, such as parts of Alaska, and thus don't capture the non-linear combination of accumulation zones on the high peaks and ablations zones in the valleys. This is indeed a shortcoming of our modelling setup and we have now addressed this in *Lines 372-375* in *Section 3.2*:

Our modelling setup also does not account for sub-grid mass balances, which can be especially relevant over mountainous regions with large sub-grid relief such as Alaska (Le Morzadec et al., 2015). Coarse grids tend to average out such tall peaks and low valleys and thus don't capture the non-linear combination of accumulation zones on the high peaks and ablation zones in the valleys.

We have also added more discussion about discrepancies in the current study and further steps for model improvement in *Lines 543-577* in *Section 4*:

The simulated ice sheet volume is well within the range of reconstructions for a rather narrow range of parameters. Small changes in parameter values can produce strongly diverging trajectories, and the emergence of multiple equilibrium states may also suggest the model's dependence on initial conditions. This poses a challenge, as many ice sheet and climate model parameters remain poorly constrained. In this context, we note that parameterizations associated with hydrofracturing and cliff instability did not impact our ice sheet trajectories. These processes have provided substantial contributions to the rapid Antarctic ice sheet retreat simulated in response to future climate projections (DeConto and Pollard, 2016), and better constraining these parameterizations is important to reduce uncertainties related to future sea level trajectories (e.g., Edwards et al., 2019). Presumably, these processes did not play an important role in our present simulations, because the climate is generally too cold, suggesting that opportunities for constraining these parameters in glacial simulations may be limited. We further note that the parameter sets which allowed for the most realistic simulation of glacial inceptions during

MIS 7-MIS 6 may not necessarily be optimal for other periods. That optimal parameter sets can depend on the period over which they are optimized, has recently been shown for a similar coupled climate ice sheet model (Bahadory et al., 2020).

Our present setup has difficulties in realistically simulating both Laurentide and Eurasian ice sheets simultaneously and generates a smaller Eurasian ice sheet compared to reconstructions, which could be a model dependent feature of LOVECLIM, given it is a T21 grid with only three levels in the atmosphere, and so could vary with the choice of the climate model used. Since we use an accelerated setup, we only conserve the freshwater flux from the ice model to LOVECLIM, which could lead to an underestimation of the oceanic circulation changes due to the lesser volume of net freshwater being dumped into the ocean. Nevertheless, there is scope of further improving the current setup. For instance, we only implement temperature and precipitation bias corrections in the current setup, and including bias corrections for radiation and ocean temperature might improve our representation of ice sheets. Future research might further improve the current setup by including the advective precipitation downscaling scheme (Bahadory and Tarasov, 2018) to account for orographic forcing, which is not captured in LOVECLIM. We are also investigating the possibilities of using a dynamical, an altitude-dependent and a $CO_2$-dependent lapse rate corrections while downscaling temperature from LOVECLIM to PSUIM. This is because the atmospheric lapse rate depends on the atmospheric $CO_2$ concentration – an effect that has not been considered so far in glacial dynamics. Furthermore, improving our basal sliding coefficient map for the NH using information of sediment sizes, instead of simply using a binary coefficient map, has the potential of further improving the simulations.

Potentially more realistic results could be obtained if the simulations were unaccelerated (which would be computationally very expensive), and from using more complex climate models that include stratification-dependent mixing in the ocean for instance. Furthermore, Glacial Isostatic Adjustment (GIA) processes captured only in comprehensive full-Earth models such as forebulges are not simulated in the ice-sheet model used here. Nevertheless, we would like to reiterate that simulating a trajectory is more difficult than conducting timeslice experiments, as climate and ice sheet components work on totally different timescales and a fine interplay of parameters can add up to very different equilibrium states. And such coupled climate-ice sheet paleo-simulations offer great opportunities for constraining parameter sets for future simulations.

*The other hole in the paper for me is pervasive in paleo ice sheet modelling: a very limited exploration of the impact of model uncertainties, given the small number of ensemble parameters and limited ensemble size. This is an exploratory work, and so arguably gets a pass with this limited ensemble, but I encourage the authors to expand their set of ensemble parameters and ensemble size in future work. And the paper needs a bit more attention to discussion of uncertainties that arise from the very limited ensemble size and the potential impact thereof.*

*The paper structure is logical. The abstract is concise and appropriate. The language is fluent, though there are instances where precision is lacking (eg "reasonably well", cf detailed commments below) as are some important (to me at least..) details about model setup.*

A.  We have now updated the table of experiments (Table 1) and added a figure in the supplementary showing all the experiments that were conducted as part of our ensemble (including those not discussed in the paper). We have also included discussions in the methods and results sections to further clarify the ensemble parameters, model uncertainties and future works. Answers to these questions are mentioned below as responses to the specific comments.

**Specific comments:**

*For a range of model parameters, the simulations capture the reconstructed evolution of global ice volume reasonably well # What does "reasonably well" mean. Be precise*

A.  Reasonably well means within the uncertainty bounds of reconstructions. We have now replaced this in *Lines 18-19* in the *Abstract*:

> For a range of model parameters, the simulations capture the evolution of global ice volume well within the range of reconstructions.

*It is demonstrated that glacial inceptions are more sensitive to orbital variations, whereas terminations from deep glacial conditions need both orbital and greenhouse gas forcings to work in unison*
*# this likely depends on your choice of fixed orbital configuration # cf Tarasov and Peltier, JGR 1997.*

A. We have clarified this to be true for our time period of consideration. This is mentioned in *Lines 19-21* in the *Abstract*:

> Over the MIS7-6 period, it is demonstrated that glacial inceptions are more sensitive to orbital variations, whereas terminations from deep glacial conditions need both orbital and greenhouse gas forcings to work in unison.

*This poses a general challenge for transient coupled climate-ice sheet modeling.# on the flip side, it poses a strong constraint opportunity, cf# Bahadory et al, cp-2020-1.pdf in TCD*

A. We agree and have added this in *Lines 24-26* in the *Abstract*:

> This poses a general challenge for transient coupled climate-ice sheet modeling, with such coupled paleo-simulations providing opportunities to constrain such parameters.

*which correspond to about 1.3 mm/year global sea level equivalent during the build-up phase.# that number is more than a factor too small for last glacial# inception if one goes by the cited LR04 stack*

A. We have now modified this to clarify that this value is an average and the changes were much higher during the LGM. This is in *Lines 36-39* in the *Section 1*:

> One of the main obstacles in simulating variability on orbital timescales is the fact that ice-sheets are slow integrators of small imbalances between ablation and accumulation, which correspond to an average of 1.3mm/year global sea level equivalent during the build-up phase but can exceed 10mm/year for instance during the Last Glacial Maximum (LGM, 21ka).

*fig 3 captions # again mixing up ensemble with ensemble run. An ensemble is a collection # of model runs.*
*fig 3 # does this show all the model runs in the non-fixed forcing ensemble you # carried out? If so, please make this clear.*
*including multi-ensemble simulations# do you mean mult-run or did you actually carry out multiple ensembles? # If so, how large was each ensemble?*

A. We use only one ensemble of multiple runs in our simulations. We also updated Table 1 to show all the ensemble runs performed in this study. While we performed a total of 50 separate experiments, we reported only 15 of them in Figure 3 that best describe the parameter sensitivities. We have now clarified this in in *Lines 293-300* in *Section 2.4:*

> Furthermore, sensitivity experiments with different GHG sensitivities ($\alpha$, Sect. 2.1) and melt parameterizations ($m$, Sect. 2.2) are run with full forcing. Generally, higher $\alpha$ leads to a stronger sensitivity to $CO_2$ concentrations, and higher values of $m$ strengthen buildup and weaken melting of ice during interglacial climates. These experiments are presented in the first row of Table 1 (1-15) and Fig. 3. Additional simulations with different combinations of acceleration ($N_A$), GHG sensitivity ($\alpha$), melt parameter ($m$), basal sliding coefficient maps over the NH ($C(x,y)$) and higher ice model resolution ($0.5 \times 0.25°$ for NH, $20 \times 20$ km polar stereographic for Antarctica) have been performed (experiments 16-50 in Table 1). The whole ensemble of simulations is presented in Fig. S3. Although we note that these experiments do not present a systematic evaluation of the full parameter space, ice sheet trajectories are consistent with and thereby support the conclusions presented in this paper.

We have also updated the text throughout the manuscript to clarify that we do not run multiple ensembles but run multiple ensemble members.

We also updated the table of experiments (Table 1):

| Expt Number | Orb Forced | GHG Forced | $N_A$ | α | $m$ (Wm$^{-2}$) | $C$ (myr$^{-1}$Pa$^{-2}$)-NH |
|---|---|---|---|---|---|---|
| 1 **(BLS)** | **Y** | **Y** | **5** | **2** | **125** | |
| 2 | **N** | **N** | 5 | 2 | 125 | |
| 3 | Y | **N** | 5 | 2 | 125 | |
| 4 | **N** | Y | 5 | 2 | 125 | Binary distribution |
| 5 | Y | Y | 5 | 2 | 125 | |
| 6 | Y | Y | 5 | **1.8** | 125 | **1.**Ocean: |
| 7 | Y | Y | 5 | **2.2** | 125 | $C(x,y)=10^{-6}$; |
| 8 | Y | Y | 5 | **2.5** | 125 | representing |
| 9 | Y | Y | 5 | **3** | 125 | deformable sediments |
| 10 | Y | Y | 5 | 2 | **80** | **2.**Land: |
| 11 | Y | Y | 5 | 2 | **100** | $C(x,y)=10^{-10}$; |
| 12 | Y | Y | 5 | 2 | **120** | representing non- |
| 13 | Y | Y | 5 | 2 | **130** | deformable rock. |
| 14 | Y | Y | 5 | 2 | **140** | |
| 15 | Y | Y | 5 | 2 | **150** | |
| *16-20* | *Y* | *Y* | *5* | *1.5* | *120,125,130,140,150* | |
| *21-24* | *Y* | *Y* | *5* | *3.5* | *80,100,120,125* | |
| *25-27* | *Y* | *Y* | *1 (30ky run)* | *2* | *110,120,130* | |
| *28-30* | *Y* | *Y* | *2 (30ky run)* | *2* | *110,120,130* | *Binary* |
| *31-33* | *Y* | *Y* | *10* | *2* | *110,130,150* | |
| *34-36* | *Y* | *Y* | *10* | *2.5* | *110,120,130* | |
| *37-38* | *Y* | *Y* | *20* | *2.5, 3* | *125* | |
| | | | | | | *Tertiary* |
| | | | | | | *1.**Ocean:* |
| | | | | | | $C(x,y)=10^{-6}$; |
| *39-41* | *Y* | *Y* | *5* | *2* | *125* | *2.1* *Land (soft tills):* |
| *42-44* | *Y* | *Y* | *5* | *2* | *150* | $C(x,y)=$**$10^{-7},10^{-8},10^{-9}$** |
| *45-47* | *Y* | *Y* | *5* | *2.5* | *125* | *over northeastern* |
| | | | | | | *North America* |
| | | | | | | *2.2* *Land (hard bed):* |
| | | | | | | $C(x,y)=10^{-10}$ |
| *High Resolution Runs:* 0.5 × 0.25° *for NH,* 20 × 20 km *polar stereographic for Antarctica* | | | | | | |
| *48-50* | *Y* | *Y* | *5* | *2* | *110,130,150* | *Binary* |

**Table 1: List of all ensemble runs performed for the study study (shown in Fig. S3). The first 15 experiments are discussed in Sect. 3.1 and shown in Fig. 3. Values in bold represent the difference from the baseline simulation (BLS, experiment number 1). $N_A$ represents the PSUIM vs LOVECLIM acceleration factor (Sect 2.3). α represents the GHG sensitivity scaling factor (Eq. 1, Sect. 2.1) and $m$ represents the constant parameter in the surface energy balance equation (Eq. 3, Sect. 2.2). $C$ represents the basal sliding coefficient map used for the NH (Eq. 7, Sect. 2.2). All experiments are run at $1 \times 0.5°$ resolution for the Northern Hemisphere and $40 \times 40$ km polar stereographic resolution for Antarctica. The experiments in italics (16-50) are not presented here but were also performed to better constrain the parameter sensitivities.**

And added a figure in the supplementary, Fig. S3, showing all the ensemble runs used in our study:

[Figure]

**Figure S3: Transient LOVECLIP ensemble simulations over MIS7 with varying GHG sensitivities (α = 1.5-3.5), energy balance parameter ($m$ = 80-150Wm$^{-2}$), basal sliding coefficient ($C$ = 10$^{-6}$-10$^{-8}$ myr$^{-1}$Pa$^{-2}$) and PSUIM-vs-LOVECLIM acceleration factor ($N_A$ = 1,2,5,10,20). The best results are obtained for α=2, $m$=125 Wm$^{-2}$, binary sliding map (ocean: $C$=10$^{-6}$ myr$^{-1}$Pa$^{-2}$ and land: $C$=10$^{-8}$ myr$^{-1}$Pa$^{-2}$) and $N_A$=5 (experiment 1 in Table 1, BLS).**

*The effect of CO2 variations with respect to the reference CO2 concentration (365ppm) on the longwave 120 radiation flux is scaled up by a factor α, to account for the low default sensitivity of ECBilt to changes in CO2 concentrations (Friedrich and Timmermann, 2020;Timmermann and Friedrich, 2016). α is determined based on transient past and future simulations.# Please provide the pCO2 ECR for alpha=2 with your setup. This would # let reader better judge how consistent this resultant sensitivity is# compared to that of IPCC grade GCMs. Also, it would be worthwhile# comparing your \alpha to that found based on 1D radiative-convective # modelling (Ramanathan et al, 1979 JGR).*

A.   Our ECS is 3.69K for CO$_2$ doubling for α=2, well within the ranges for LOVECLIM only simulation reported in Friedrich et al. (2016) and Friedrich and Timmermann (2020). We have now mentioned this in *Line 129* in the *Section 2.1*:

> For reference, the equilibrium climate sensitivity for CO$_2$ doubling is 3.69K for α of 2.

*2.2 PSUIM surface mass balance description, eq 1 and 2 # on what timestep is this carried out? If longer than 1 hour # (presumably), what accounting is there for diurnal variations?*

A.   The surface mass balance is calculated at 3-hourly timesteps. The monthly data is interpolated to daily values using a weighted average of values across the adjacent months and then a sinusoidal cycle with max temperatures at 1400 and minimum at 0200, with peak-to-peak amplitude of 10°C, is superimposed to account for daily variations.
We have now clarified this in *Lines 168-173* in the *Section 2.2*:

> This surface mass balance is calculated at timesteps of 3 hours. Monthly surface air temperature ($T$) and surface incoming shortwave radiation ($Q$) (obtained from LOVECLIM in the current setup, discussed further in Sect. 2.3) are interpolated into sub-daily values in two steps. Firstly, the monthly values are interpolated to daily values using a weighted averaging of the values across two adjacent months. Next, a sinusoidal cycle with max temperature at 1400 and minimum at 0200, with a peak-to-peak amplitude of 10°C, is superimposed on the daily data to account for diurnal variations.

*after eq 4 : with r = maxJ0, min[1, (T\* + 3)/3] = maxJ0, min[1, (T\* + 3)/3]J0, min[1, (T\* + 3)/3][1, (T\* + 3)/3]\* + 3)/3]*
*# Based on my on examination of ice sheet model horizontal basal # temperature between along flow adjacent grid cells (which*
*provides # logical upper bound for the transition range), 3 C is a wide # transition range for warm based sliding. How is this*
*justified?*

A.  The 3°C range was chosen during model development of PSUIM to improve the realism of modern ice thicknesses and
    flow over certain regions of Antarctica, and avoid ubiquitous frozen beds over the Transantarctic for instance (Pollard and
    DeConto, 2012). Instead of using temperatures between adjacent along-flow grid cells as upper bounds, the authors use
    this 3°C temperature range to represent the sub-grid variations in basal temperatures within a single grid cell, due to small-
    scale variations of bed properties, roughness and topography.

*For the NH, a binary sliding coefficient map ... low sliding over present-day land (C(xJ0, min[1, (T\* + 3)/3], y) = ...*
*representing non-deformable rock).) = ... representing non-deformable rock). # Much of Southern Canada and Northern USA*
*(regions of glacial ice # cover) is covered by tills, not hard beds and this can significantly # influence ice sheet evolution (eg*
*Tarasov and Peltier, 2004 # QSR). How do you justify making all this hard bedded?*

A.  While we apply only a binary basal sliding coefficient map in the current study, we did try a tertiary map (with intermediate
    sliding over the northeastern North America) and the ice sheet evolutions were similar. Given this is the first study using
    this coupled setup, we presented results using the simple binary sliding coefficient map distribution. We are currently
    working on using an improved sliding map based on the sediment size data. We have added this as discussion of future
    work in in *Lines 567-569* in the *Section 4*:

> Furthermore, improving our basal sliding coefficient map for the NH using information of sediment sizes, instead of simply
> using a binary coefficient map, has the potential of further improving the simulations.

*Preliminary experiments (not shown) with different acceleration factors suggest that model results do not change significantly*
*when N <= 5. # Please be more precise by what "significantly" means.*

A.  We wanted to convey that the simulated ice volume evolutions were similar over these low acceleration experiments. We
    have now replaced this in *Lines 215-217* in the *Section 2.3*:

> Preliminary experiments (not shown) with different acceleration factors suggest that the simulated ice sheet evolution is
> relatively insensitive to $N_A$ for $N_A \leq 5$. Therefore, $N_A = 5$ is used for the simulations presented in this paper, providing a
> good compromise between the objective to simulate realistic ice sheet evolution and computational efficiency.

*Furthermore, for surface temperature T, a lapse-rate correction of 8 °C km−1 is applied to account for differences between*
*LOVECLIM orography and PSUIM topography and precipitation is multiplied by a Clausius–Clapeyron factor of 2^.. with*
*$\Delta T$ being the temperature lapse-rate correction, to account for the elevation desertification effect (DeConto and*
*Pollard, 2016).# How do you justify using a lapse rate that is inconsistent with# the lapse rate LOVECLIM uses internally?*
*For future work, I would # strongly advise inclusion of orographic forcing given the impact# thereof missed in a coarse grid*
*EMIC (cf eg Bahadory# and Tarasov, GMD 2018)*

A.  Firstly, both the models LOVECLIM and PSUIM, and the parameterizations therein, were developed independently.
    Secondly, we use a higher lapse rate in PSUIM because we are more interested over the high latitudes in terms of ice sheet
    evolution. These regions are also drier and would have a higher lapse rate compared to the environmental lapse rate used
    in LOVECLIM. We are currently investigating the possibilities of using a dynamic lapse rate following Roche et al. (2014)
    and Bahadory and Tarasov (2018), an altitude dependent lapse rate as in Colleoni and Liakka (2020), and a $CO_2$ dependent
    lapse rate. We have added more discussion around this in *Lines 219-248* in *Section 2.3*:

> PSUIM uses surface air temperature ($T$), precipitation ($P$), solar radiation ($Q$), and ocean temperature at 400m depth ($T_o$)
> as inputs from LOVECLIM. These are downscaled using a bilinear interpolation approach. The surface temperature and

precipitation outputs from LOVECLIM which are used for the PSUIM surface mass balance are bias-corrected in the coupler, following Pollard and DeConto (2012b), Heinemann et al. (2014) and Tigchelaar et al. (2018).

$$T(t) = T_{LC}(t) + T_{obs} - T_{LC,PD} \tag{8}$$

$$P(t) = P_{LC}(t) \times P_{obs}/P_{LC,PD} \tag{9}$$

where $T$ is monthly surface air temperature and $P$ is monthly precipitation forcing from LOVECLIM at timestep $t$. Subscripts '$LC$', '$obs$' and '$LC,PD$' refer to LOVECLIM chunk output, observed present day climatology, and LOVECLIM present day control run, respectively. The observed present day climatology is obtained from the European Centre for Medium-Range Weather Forecasts reanalysis dataset, ERA-40 (Uppala et al., 2005). These LOVECLIM biases are calculated for PD simulations using an LGM bathymetry. We did compare the biases between using a PD or LGM bathymetry, and while there were regional differences, the large-scale structure was found to be similar (not shown). The annual mean of the monthly mean bias correction terms $T_{obs} - T_{LC,PD}$ and $P_{obs}/P_{LC,PD}$ are presented in Fig. S1. Temperature biases in LOVECLIM for boreal summer (JJA) and austral summer (DJF) are shown in Fig. S2 for reference, since summer temperatures are more crucial for ice sheet growth and decay. Furthermore, a lapse-rate correction of 8°C km$^{-1}$ is applied to account for differences between LOVECLIM orography and PSUIM topography for the interpolated temperature, $T(t)$, and precipitation is multiplied by a Clausius–Clapeyron factor of $2^{\frac{-\gamma.\Delta H}{10°C}}$, with $\gamma.\Delta H$ being the temperature lapse-rate correction, to account for the elevation desertification effect (DeConto and Pollard, 2016):

$$T_{PSUIM}(t) = T_{interp}(t) - \gamma.\Delta H \tag{10}$$

$$P_{PSUIM}(t) = P_{interp}(t) \times 2^{\frac{-\gamma.\Delta H}{10°C}} \tag{11}$$

where $T_{PSUIM}$ and $P_{PSUIM}$ are the final temperature and precipitation inputs for PSUIM, $T_{interp}$ and $P_{interp}$ are bias corrected LOVECLIM temperature ($T$, Eq. 8) and precipitation ($P$, Eq. 9) interpolated to PSUIM resolution, $\gamma$ is the lapse rate (8°C km$^{-1}$), and $\Delta H$ is the altitude difference between PSUIM grids and the corresponding LOVECLIM grid. Colleoni and Liakka (2020) used a similar fixed atmospheric lapse rate correction during downscaling temperature to their ice model, GRISLI, with $\gamma$ as 3.3°C km$^{-1}$ for annual mean and 4.1°C km$^{-1}$ for summer mean. And they reported slightly smaller ice sheets on using an elevation dependent lapse rate, going all the way up to 7.9°C km$^{-1}$. Instead of using a fixed value of $\gamma$, both Roche et al. (2014) and Bahadory and Tarasov (2018) used a dynamic lapse rate, where $\gamma$ is estimated locally for the ice model grids in each LOVECLIM grid. Moreover, the lapse rate also depends on the atmospheric $CO_2$ concentration. Such dynamic lapse rate corrections are not implemented in the current setup, and neither is the advective precipitation downscaling scheme of Bahadory and Tarasov (2018).

And have added these as possibilities of further improvement in *Lines 561-569* in *Section 4*:

. Nevertheless, there is scope of further improving the current setup. For instance, we only implement temperature and precipitation bias corrections in the current setup, and including bias corrections for radiation and ocean temperature might improve our representation of ice sheets. Future research might further improve the current setup by including the advective precipitation downscaling scheme (Bahadory and Tarasov, 2018) to account for orographic forcing, which is not captured in LOVECLIM. We are also investigating the possibilities of using a dynamical, an altitude-dependent and a $CO_2$-dependent lapse rate corrections while downscaling temperature from LOVECLIM to PSUIM. This is because the atmospheric lapse rate depends on the atmospheric $CO_2$ concentration – an effect that has not been considered so far in glacial dynamics. Furthermore, improving our basal sliding coefficient map for the NH using information of sediment sizes, instead of simply using a binary coefficient map, has the potential of further improving the simulations.

*Basal melting and liquid runoff from PSUIM is discharged via LOVECLIM's runoff masks in both hemispheres; # do these masks account for changing topography? And if so, what # accounting is there for critical subgrid gateways for southern # drainage from the NA North American) ice complex (cf eg Tarasov and # Peltier, QSR 2006).*

A. The net meltwater in PSUIM is dynamically routed based on the actual topography. The current setup does not account for the subgrid pathways needed for southward drainage from the North American ice sheets. We have clarified this in *Lines 263-266* in *Section 2.3*:

The total meltwater from basal melting and liquid runoff in PSUIM is dynamically routed based on PSUIM topography till it reaches the ocean or the domain edge, and then is dumped into the nearest ocean grid point in LOVECLIM. The calving flux is channeled into CLIO's iceberg model (Schloesser et al., 2019;Jongma et al., 2009) in the Southern Hemisphere (SH) and as an iceberg melt flux (freshwater flux and heat flux) in the NH (Schloesser et al., 2019).

*Increasing the value of m (Eq. (1)) # as a reader, it is a pain to flip back 5 pages to find out what m # is, please add a few descriptive words (surface energy offset term # or some such) ditto for \alpha*

A. Done.

*3.2 Ice sheet evolution # this section would be strengthened with more contact with the # (albeit limited) glacial geological litterature. The key relevant # data are Late Pleistocene glacial limits. Does your model respect # them everywhere? If not, what are the main discrepancies? The only # regions I see that could be at issue are your Alaskan incursion and # Northern Siberia.*

A. While we could not find many reconstructions over this period, we have now added discussions comparing our simulations with other modelling and reconstruction studies. Some excerpts of these are mentioned under from *Lines 344-376* in *Section 3.2*:

In the context of previous modelling studies and geological records over this MIS 7-6 period, our ice sheet distribution at MIS 7c (212ka, Fig. 4g and 219.5ka, Fig. S7) is very similar to that reported in Colleoni and Liakka (2020). However, we simulate a stronger inception compared to that of Colleoni et al. (2014b) over the corresponding 236-230ka period. They also reported a bifurcated but connected North American ice sheet at MIS 6 (157ka) from both their control (100km) and high resolution (40km) experiments. Our simulation results in separate Laurentide and Cordilleran ice sheets but generates neither a Eurasian nor a Siberian ice sheet, albeit at 170ka. On a side note, our North American ice sheet distribution at 180ka (Fig. 7) is closer to that of Colleoni and Liakka (2020) at 157ka. Studies of NH reconstructions during MIS 6 such as Svendsen et al. (2004), over 160-140ka, Rohling et al. (2017), around 140ka, and Batchelor et al. (2019), over 190-132ka, have all reported glacial geological records to indicate a larger extent of the Eurasian ice sheet at MIS 6 glacial maximum compared to the LGM, while our simulations only show a persistent Fenno-Scandian ice sheet and a relatively small Eurasian ice sheet at 170ka. More recently, Zhang et al. (2020) reported the existence of a Northeast Siberia-Beringian ice sheet at MIS 6e (190-180ka) using NorESM-PISM simulations validated by North Pacific geological records. However, our model does not simulate any ice over Alaska, Beringia and northeast Siberia over MIS 7-6.

Our model's difficulty in simulating the Eurasian ice sheet can be attributed to the competition between Laurentide and Eurasian ice sheet growth, which makes it arduous to realistically simulate them simultaneously alongside generating the right atmospheric patterns. Some previous studies have suggested that teleconnections from stationary wave patterns induced by a large Laurentide ice sheet could lead to warming over Europe and influence Eurasian ice sheet evolution (Roe and Lindzen, 2001;Ullman et al., 2014). The Laurentide building up first in our simulations could have changed the storm tracks and dried out Eurasia. It is also worth reiterating that LOVECLIM has a coarse T21 grid with a simple 3-layered atmosphere. While the circulation changes reported here maybe model dependent, Lofverstrom and Liakka (2018) reported that at least a T42 grid was needed in their atmospheric model (CAM3) to generate a Eurasian ice sheet using SICOPOLIS, albeit for the LGM. They attribute this discrepancy to lapse rate induced warming due to reduced and smoother topography and higher cloudiness leading to increased re-emitted longwave radiation towards the surface. These teleconnection patterns are further discussed in Sect. 3.6. Our LOVECLIM setup also uses a fixed lapse rate for downscaling LOVECLIM surface temperatures (Eq. 10 and 11), while both Roche et al. (2014) and Bahadory and Tarasov (2018) used a dynamic lapse rate, which is estimated locally for the ice model grids in each LOVECLIM grid. Bahadory and Tarasov (2018) reported ice thickness differences up to 1km on using the dynamic lapse rate scheme compared to a fixed 6.5˚Ckm$^{-1}$. Nevertheless, for runaway trajectories, our model can build up a Eurasian ice sheet for ice volumes greater than -200m SLE once the Laurentide growth slows down (not shown). Our modelling setup also does not account for sub-grid mass balances, which can be especially relevant over mountainous regions with large sub-grid relief such as Alaska (Le Morzadec et al., 2015). Coarse grids tend to average out tall peaks and low valleys and thus don't capture the non-linear combination of

accumulation zones on the high peaks and ablation zones in the valleys. These shortcomings could explain the lack of Eurasian, Siberian and Beringian ice sheets in our simulations.

*the glaciation 235 into MIS 6 is delayed by ~3ky (191ka instead of 194ka). # Do you really believe that temporaly uncertainty in inferred# sealevel is < 3 kyr that far back?*

A.   We agree with Lev on this and have now added this to our discussion of results in *Lines 309-313* in *Section 3.1*:

The model captures the overall trajectory of ice volume evolution reasonably well. Specifically, the model stays within the uncertainty range for the extreme glaciation-deglaciation event of MIS 7e-7d-7c. Larger differences only exist as the glaciation into MIS 6 is delayed by ~3ky in the simulation (191ka instead of 194ka). A possible explanation for this discrepancy may be related to the temporal uncertainty in reconstructions themselves, since a similar lag occurs in other modeling studies (e.g., Ganopolski and Calov (2011); Ganopolski and Brovkin (2017).

*After a relatively stable interglacial state till MIS 7a, the system moves into the next glacial and reaches a glacial equilibrium state. # This description does not accurately reflect your figure 3, I see no # sign of a "glacial equilibrium"*
*...Batchelor et al. (2019), have suggested a larger Eurasian ice sheet over the MIS 6 period (160-140ka), # "suggested" does not accurately nor precisely reflect the # inferences. Be more accurate: eg glacial geological record indicates # that the asynchronous maximal MIS 6 ice margins are outside of MIS 2 # ice margins.*

A.   We have now removed the phrase about "glacial equilibrium" and have clarified MIS 6 ice sheet reconstructions in *Lines 350-356* in *Section 3.2*:

Studies of NH reconstructions during MIS 6 such as Svendsen et al. (2004), over 160-140ka, Rohling et al. (2017), around 140ka, and Batchelor et al. (2019), over 190-132ka, have all reported glacial geological records to indicate a larger extent of the Eurasian ice sheet at MIS 6 glacial maximum compared to the LGM, while our simulations only show a persistent Fenno-Scandian ice sheet and a relatively small Eurasian ice sheet at 170ka. More recently, Zhang et al. (2020) reported the existence of a Northeast Siberia-Beringian ice sheet at MIS 6e (190-180ka) using NorESM-PISM simulations validated by North Pacific geological records. However, our model does not simulate any ice over Alaska, Beringia and northeast Siberia over MIS 7-6.

*# leading to temperatures low enough (Fig. 6d) to avoid ablation even if # the Laurentide extends equatorward There is always seasonal ablation on an northern ice sheet. Be more precise.*

A.   We have now rephrased these in *Lines 440-442* in *Section 3.4*:

This can be attributed to the low $CO_2$ value (<200ppmv) leading to lower temperatures (Fig. 6d) and reduced ablation even if the Laurentide extends equatorward (Fig. 6g). Furthermore, the southern extent of the Laurentide can lead to changes in circulation patterns that can alter the SMB (discussed in Sect. 3.6).

*Figure 7: # makes it a lot easier for the reader if subplots have descriptive # headings on the plot. Having to visually jump between each subplot and a large # caption disrupts reader assimilation of the plots.*

A.   Done.

*Fig 7 caption two ensembles of # do you mean two ensemble members?*

A.   Yes. We have updated ensembles to ensemble members throughout the study.

*Fig 7f-I # I find the colour scheme has insufficient and distorting colour range. Eg # for 7h the 0.3:0.5 colour is just a shade darker than the -0.3:-0.1 range # colour. Furthermore, it makes no sense that the plot has regions where # these colour border each other without any intermediate ranges showing.*

A.  We have now used a better colormap to plot negatives in blue and positives in red throughout the manuscript.

*Fig 7: # I am a bit confused why there is such limited glaciation east of the # Canadian Cordillera, given the northwesterly (and therefore relatively colder) # absolute winds and rainfall anomalies that match (within the colour # scheme) other sectors with significant ice cover. Is this due to # the temperature bias correction or limited rainful or ? On that note, # a short discussion on the impact of the bias correction would aid # interpretation of its role in your results.*

A.  We suspect the reduced glaciation east of the Canadian Cordillera because of low net precipitation (not anomalies). The precipitation bias over this region is almost 1 (Fig. S1). While Fig. 7 shows the *anomalies* in mass balance terms with reference to those at 240ka, the figure underneath (Fig. R1) shows the *absolute* values of these mass balance terms. Fig. R1 (h) and (i) show that the precipitation just east of the Cordilleran is very low. To make these patterns clearer, we have now added a figure in the supplementary showing the initial patterns of the mass balance variables at 240ka for comparison (Fig. S8 now) with the anomalies presented in Fig. 7 and Fig. S9.

[Figure]

**Figure R1: Bifurcation of the system at 180ka while transitioning into MIS 6 over Laurentide. (a)** Sea level reconstruction (m) and 95% confidence interval of Spratt and Lisiecki (2016) (brown). Total ice volume (in terms of SLE, m) from two ensemble members of LOVECLIP, one that leads to a stable glacial inception (blue; $\alpha$=2, $m$=125 Wm$^{-2}$) and another into a runaway glaciation (black; $\alpha$=2, $m$=130 Wm$^{-2}$). Climate and ice sheet variables at 180ka from the stable glaciation on the left column (b, d, f and h) and runaway glaciation on the right (c, e, g and i). **(b,c)** Basal ice velocity (solid colors, my$^{-1}$) overlaid with ice thickness (colored contours, km) and the grounding line (solid green lines). **(d,e)** Surface temperature (˚C) overlaid with wind vectors at 800hPa (ms$^{-1}$). **(f,g)** Net mass balance (my$^{-1}$) overlaid with winds (ms$^{-1}$). **(h,i)** Net accumulation (my$^{-1}$) overlaid with winds (ms$^{-1}$). The purple contours in (d) to (i) mark the boundaries of the ice sheets from each run (stable for left and runaway for right).

*this behavior is reminiscent of a saddle node bifurcation*

*We find that small changes in the Laurentide's ice distribution for similar total ice volumes can lead to a saddle node 400 bifurcation of the system # which is correct? Have you shown this to be a saddle node bifurcation # or is this reminiscent of a saddle node bifurcation?*

A.  We apologize for this confusion. We see small differences in ice sheet distributions reminiscent of a saddle node bifurcation. We have now updated this in the text in *Lines 536-538* in *Section 4*:

> We find that small changes in the Laurentide's ice distribution for similar total ice volumes reminiscent of a saddle node bifurcation, which in turn determines whether the coupled trajectory will follow a deglaciation or a runaway glaciation pathway in response to the combination of forcings.

*Also, the stationary wave feedback reported here 410 could be a model dependent feature of LOVECLIM, given it has only three atmospheric levels # and LOVECLIM is run at a relatively coarse T21, while the*
*# litterature indicates that at least T42 is needed to avoid major # resolution sensitivity of the eddy driven jet (eg Lofverstrom and # Liakka, 2018).*

A.  We thank Lev for this suggestion. We have now discussed and expanded on this in *Lines 358-368* in *Section 3.2*:

> Our model's difficulty in simulating the Eurasian ice sheet can be attributed to the competition between Laurentide and Eurasian ice sheet growth, which makes it arduous to realistically simulate them simultaneously alongside generating the right atmospheric patterns. Some previous studies have suggested that teleconnections from stationary wave patterns induced by a large Laurentide ice sheet could lead to warming over Europe and influence Eurasian ice sheet evolution (Roe and Lindzen, 2001;Ullman et al., 2014). The Laurentide building up first in our simulations could have changed the storm tracks and dried out Eurasia. It is also worth reiterating that LOVECLIM has a coarse T21 grid with a simple 3-layered atmosphere. While the circulation changes reported here maybe model dependent, Lofverstrom and Liakka (2018) reported that at least a T42 grid was needed in their atmospheric model (CAM3) to generate a Eurasian ice sheet using SICOPOLIS, albeit for the LGM. They attribute this discrepancy to lapse rate induced warming due to reduced and smoother topography and higher cloudiness leading to increased re-emitted longwave radiation towards the surface. These teleconnection patterns are further discussed in Sect. 3.6.

And in *Lines 520-522* in *Section 3.6*:

> As mentioned earlier in Sect. 3.2, it is important to acknowledge the low horizontal and vertical resolutions of LOVECLIM's atmosphere, which could mean the circulation changes reported here to be model dependent.

*Results also suggest that our coupled simulations are realistic over a narrow range of parameters # what does "realistic" mean? Again, be precise*

A.  We meant 'realistic' simulations to have good agreement with the reconstructions. We have clarified this in *Lines 543-545* in *Section 4*:

> The simulated ice sheet volume is well within the range of reconstructions for a rather narrow range of parameters. Small changes in parameter values can produce strongly diverging trajectories, and the emergence of multiple equilibrium states may also suggest the model's dependence on initial conditions.

*is more difficult than conducting timeslice experiments # I would say much more difficult and therefore offers much more # self-constraint*

A.  Agreed. We have added in *Lines 574-577* in *Section 4*:

> Nevertheless, we would like to reiterate that simulating a trajectory is more difficult than conducting timeslice experiments, as climate and ice sheet components work on totally different timescales and a fine interplay of parameters can add up to

very different equilibrium states. And such coupled climate-ice sheet paleo-simulations offer great opportunities for constraining parameter sets for future simulations.

*Fig S1 # summer (JJA for NH and DJF for SH) temperature is much more critical # for ice sheet growth than mean annual temperature, given surface mass-balance # dependencies, so please add these plots.*

A.  We have now added the summer temperature biases in Fig. S2 in the supplementary.

[Figure]

**Figure S2: Seasonal biases in surface temperature (K) from LOVECLIM for (a) JJA, and (b) DJF.**

**Reference:**

[revised manuscript text omitted]